# FM-SIREN & FM-FINER: Nyquist-Informed Frequency Multiplier for Implicit Neural Representation with Periodic Activation

## Abstract

Existing periodic activation-based implicit neural representation (INR) networks, such as SIREN and FINER, suffer from hidden feature redundancy, where neurons within a layer capture overlapping frequency components due to the use of a fixed frequency multiplier. This redundancy limits the expressive capacity of multilayer perceptrons (MLPs). Drawing inspiration from classical signal processing methods such as the Discrete Sine Transform (DST), we propose FM-SIREN and FM-FINER, which assign Nyquist-informed, neuron-specific frequency multipliers to periodic activations. Unlike existing approaches, our design introduces frequency diversity without requiring hyperparameter tuning or additional network depth. This simple yet principled modification reduces the redundancy of features by nearly $50\%$ and consistently improves signal reconstruction across diverse INR tasks, including fitting 1D audio, 2D image, 3D shape, and synthesis of neural radiance fields (NeRF), outperforming their baseline counterparts while maintaining efficiency.

## 1 Introduction

Signal representation often depends on selecting an appropriate basis tailored to the application, where each basis has its own pros and cons. The commonly used standard basis, composed of $N$ ordered permutations of one-hot vectors that form the identity matrix, leads to a coordinate-value description known as an *explicit representation*, which is simple to comprehend and visualize (Nordberg, 1992). While straightforward, this representation necessitates storage that scales directly with signal size, which can limit efficiency in applications requiring fast processing, such as those involving linear convolutions or cross-correlation (e.g., Fourier Transform) (Hayes, 1996). To address this, alternative basis functions are often used to represent signals more compactly. For example, Cosine, Sine, and Fourier Transforms decompose signals into periodic components, enabling efficient processing and offering compact representations. These transformations remain within the domain of *explicit representation* (Bracewell & Kahn, 1966), as the basis functions are analytically defined and the coefficients are directly interpretable. In recent years, such bases have also been adopted within *implicit neural representations* as activation functions (Essakine et al., 2024).

Various methods for explicit signal representation have been proposed by researchers, including the Discrete Sine Transform (DST) (Ahmed et al., 2006), Discrete Fourier Transform (DFT) (Brigham, 1988), Discrete Wavelet Transform (DWT) (Daubechies, 1992), and the fractional Fourier transform (FrFT) (Lin, 1999). Although these classical techniques are computationally efficient, they are inherently limited to producing linear combinations of fixed basis functions, which may restrict their expressive capacity to represent complex or non-stationary signals. Interestingly, many of these basis functions are now reused in multilayer perceptrons (MLPs) as activation functions in modern implicit neural representations. For instance, the Sinusoidal Representation Network (SIREN) uses sinusoids (Sitzmann et al., 2020), Wavelet Implicit Neural Representations (WIRE) uses wavelets (Saragadam et al., 2023), and Flexible spectral-bias tuning in Implicit NEural Representation (FINER) employs chirp-based activations (Liu et al., 2024).

MLPs offer the significant advantage of modeling hierarchical signal components through compositions of non-linear neurons (Popescu et al., 2009). However, they are inefficient at learning implicit representations due to their reliance on iterative optimization. Training even small MLPs requires

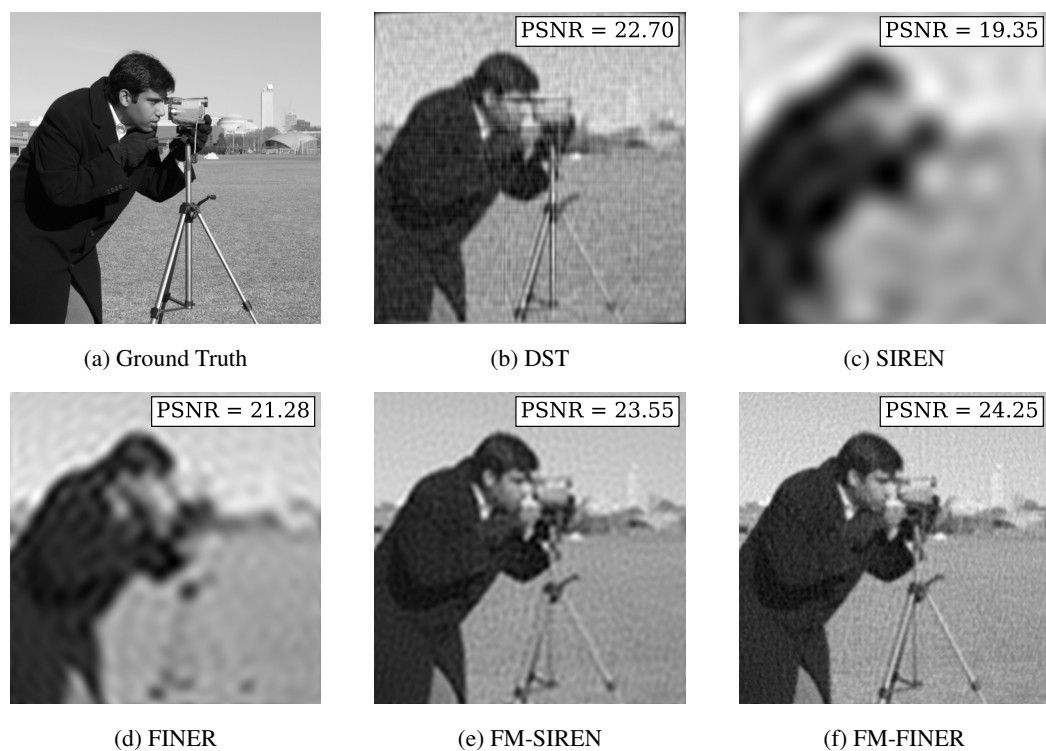

(a) Ground Truth      (b) DST      (c) SIREN

(d) FINER      (e) FM-SIREN      (f) FM-FINER

Figure 1: Comparison of reconstruction quality using different methods for the cameraman image from USC Signal and Image Processing Institute (1973) database. Subfigures (c–f) show INR-based reconstructions with single-layer networks, designed to match the linear DST baseline in (b). All models were trained for 500 epochs using the Adam optimizer. FM-SIREN and FM-FINER yield the highest PSNR values, outperforming both SIREN and FINER as well as the classical DST.

several seconds, whereas computing a DST is virtually instantaneous. This inefficiency likely stems from the limited design of existing activation functions, even though a single-layer MLP structurally resembles the linear combination produced by classical explicit representation methods. Figure 1(b–d) compares the reconstruction results of single-layer MLPs with SIREN and FINER activations against DST reconstruction, all using the same number of parameters (2048). One can observe that SIREN performs significantly worse and that FINER also falls short despite its frequency-variable activation, which, in principle, should offer higher capacity (i.e., bandwidth) (Collins & Atkins, 1999). These results indicate that the linear layers of state-of-the-art (SOTA) implicit neural representations still lack the expressive capacity of classical explicit methods despite using the same bases functions as activation.

In this paper, we address the problem of limited linear layer capacity in INRs by studying frequency diversity within the same layer. Our approach is inspired by the orthogonality principle underlying classical transforms (Tolimieri, 1984). In explicit representations, coefficients are obtained through inner products with orthogonal basis functions, ensuring uncorrelated components. This contrasts with conventional INRs, where network parameters are optimized while the frequency multiplier is kept fixed for all activations in a layer. To achieve good performance, researchers typically increase the number of layers and parameters while relying on random initialization, which often results in correlated and redundant hidden features. We hypothesize that explicitly inducing frequency diversity in a single layer can significantly enhance its representational capacity. To this end, we propose activation functions that induce frequency diversity among neurons within a single layer, informed by the Nyquist sampling theorem (Por et al., 2019). Figures 1e and 1f present reconstruction results of our proposed methods: Frequency Multiplier-SIREN (FM-SIREN) and Frequency Multiplier-FINER (FM-FINER). Our results show consistent improvements over SIREN and FINER, and even outperform the classical DST.

Our contributions can be summarized as follows: (1) we propose a principled scheme for designing frequency multipliers in periodic activation functions of INRs, informed by the Nyquist sampling theorem, which increases frequency diversity, eliminates the need for hyperparameter tuning, and preserves efficiency without increasing network width or depth; (2) we introduce two new architectures, FM-SIREN and FM-FINER, that employ our Nyquist-informed multipliers and reduce hidden feature redundancy by $49.92\%$ and $50.43\%$, respectively; (3) we establish a connection between classical linear signal reconstruction and modern MLP-based INRs, providing theoretical insight into how Nyquist frequency limits can be incorporated into implicit neural representations; and (4) we present comprehensive experiments on 1D audio, 2D image and 3D volume reconstructions, and NeRF synthesis, demonstrating that our models consistently outperform the baselines.

## 2 RELATED WORK

Different INR techniques have been presented in the literature. Given an input coordinate $x$, the INR network learns the mapping of $f(x)$, which is represented by a standard basis. Hence, the INR network learns the implicit mapping. In this section, we review several INR techniques found in the literature.

**SIREN (Sitzmann et al., 2020).** SIREN introduces sine activations with a frequency hyperparameter $\omega$, enabling smooth function fitting and stable training. While effective for continuous signals, its fixed frequency multiplier limits flexibility, reducing accuracy on tasks with diverse frequency details. The SIREN activation function is expressed as $\sigma(x) = \sin(\omega x)$.

**GAUSS (Ramasinghe & Lucey, 2022).** The Gauss activation provides smooth, localized representations useful for denoising and signal reconstruction, but its non-periodicity limits modeling of high-frequency details. The Gauss activation function is expressed as $\sigma(x) = e^{-(sx)^2}$.

**WIRE (Saragadam et al., 2023).** WIRE employs the Gabor wavelet activation (Lee, 1996), which combines sinusoidal and Gaussian components to represent localized frequencies. This design enables effective modeling of high-frequency details in spatially localized regions, but its complex-valued formulation increases computational cost and makes training and inference more expensive. The Gabor wavelet is expressed as $\sigma(x) = e^{-(sx)^2 + i\omega x}$.

**FINER (Liu et al., 2024).** FINER modifies SIREN by using a frequency-variable activation, enabling representation of a broader frequency spectrum while retaining SIREN's advantages, making it more capable to capture broader range of frequencies. However, it is sensitive to bias initialization and adds training complexity. This activation function is expressed as $\sigma(x) = \sin(\omega(|x| + 1)x)$.

**Positional Encoding (Tancik et al., 2020).** Positional encoding (PE) maps inputs into a higher-dimensional space using sinusoidal functions as $\gamma(x) = [\sin(2^0 \pi x), \cos(2^0 \pi x), \ldots, \sin(2^{L-1} \pi x), \cos(2^{L-1} \pi x)]$ to mitigate the spectral bias of MLPs. While it enriches representations with multiple frequency components, it is sensitive to the choice of $L$ and it increases input dimensionality, raising the overall number of parameters in the network.

Existing INR models generally lack guarantees of frequency diversity, relying instead on iterative optimization to mitigate frequency bias. They also do not ensure orthogonality among hidden features within the same layer, in contrast to classical bases such as those in the DST. Recent works propose higher-level frameworks: **MIRE** (Jayasundara et al., 2025) employs a search-based algorithm to select activation functions per layer from a predefined dictionary; **Neural Experts** (Ben-Shabat et al., 2024) leverages large mixture-of-experts models (Masoudnia & Ebrahimpour, 2014) composed of smaller subnetworks; and **Fourier Reparameterization** (Shi et al., 2024) reparameterizes the weights of MLPs with a fixed Fourier basis to alleviate spectral bias. In contrast, our work focuses on the principled design of activation functions, and we therefore compare FM-SIREN and FM-FINER against other activation-based INR approaches in the following sections.

## 3 BACKGROUND

In this section, we provide the necessary background on signal reconstruction using periodic basis functions, the Nyquist sampling theorem for proper sampling, and the extension of these concepts to signal reconstruction with MLPs.

### 3.1 LINEAR SIGNAL RECONSTRUCTION

According to Stearns & Hush (1990), let a desired function $f(x)$ be approximated as closely as possible by another function $f^*(c, x) = f^*(x)$, where $c$ denotes a set of adjustable parameters used to minimize the error between the two functions. The approximation $f^*(x)$ is expressed as a linear combination of a set of periodic bases with different frequencies $[\phi_0(x), \phi_1(x), \ldots, \phi_{M-1}(x)]$, with the corresponding amplitude coefficients $[c_0, c_1, \ldots, c_{M-1}]$. Formally, $f^*(x)$ can be written as:

$$f^*(c, x) = f^*(x) = \sum_{m=0}^{M-1} c_m \phi_m(x) \tag{1}$$

The optimal approximation $f^*(x)$ is obtained by minimizing the least squares error between $f^*(x)$ and $f(x)$. A central property in this formulation is orthogonality of the basis functions, which ensures that each coefficient can be computed independently of the others. In particular, when the bases $[\phi_0(x), \phi_1(x), \ldots, \phi_{M-1}(x)]$ correspond to distinct frequencies, the coefficient $c_m$ is computed as:

$$c_m = \frac{\sum_{n=0}^{N-1} f_n(x)\, \phi_{mn}(x)}{\sum_{n=0}^{N-1} \phi_{mn}^2(x)}, \quad m = 0, 1, \ldots, M-1. \tag{2}$$

The linear independence due to the orthogonality of the basis functions is what makes classical transforms efficient and non-redundant, a property that conventional MLP activations fail to guarantee, leading to correlated hidden features.

### 3.2 NYQUIST FREQUENCY

The Nyquist frequency specifies the maximum frequency that can be captured without aliasing when sampling a continuous-time signal using periodic basis functions. According to the Nyquist sampling theorem, the sampling frequency must be at least twice the highest frequency component present in the signal to ensure a perfect reconstruction (Shannon, 2006). Sampling below this threshold causes frequency components to overlap in the spectral domain, resulting in distortion and loss of information. Formally, the Nyquist frequency is defined as:

$$f_{\text{Nyquist}} = \frac{f_{\text{s}}}{2}, \quad f_{\text{s}} \geq f_{\text{max}} \tag{3}$$

where $f_{\text{s}}$ is the sampling frequency and $f_{\text{max}}$ denotes the maximum frequency present in the signal. For example, if a signal contains frequency components up to 1 kHz, the sampling frequency must be at least 2 kHz, corresponding to a Nyquist frequency of 1 kHz. For higher-dimensional signals, such as 2D images and 3D volumes, each dimension has its own Nyquist frequency.

### 3.3 MLP RELEVANCE TO LINEAR SIGNAL RECONSTRUCTION

MLPs are composed of sequential linear layers, which can be interpreted as reconstructing linear and hierarchical embeddings. For simplicity, consider a single hidden linear layer where both the input $\mathbf{x}$ and the output $\mathbf{y}$ are one-dimensional. The output $\mathbf{y}$ of this layer with respect to the input x can be written as:

$$y_n = \sum_{m=0}^{M-1} w_m^{\text{out}} \phi_m\left(w_m^{\text{in}} x_n\right) \tag{4}$$

where $w_m^{\text{in}}$, $w_m^{\text{out}}$, and $\phi_m$ denote the input weights, output weights, and activation function, respectively. This representation resembles the linear reconstruction in Equation 1. However, in classical signal reconstruction, the basis functions $\phi_m$ correspond to distinct frequencies, avoiding redundancy. In contrast, conventional INR settings typically assign the same activation function across all neurons in a layer, effectively creating frequency replicas. As a result, the optimizer must implicitly enforce orthogonality through weight updates, which is not guaranteed due to the highly non-convex nature of MLPs. This limitation is evident in models such as SIREN and FINER, where hidden embeddings often remain correlated, reducing the overall capacity, as shown in Figure 1.

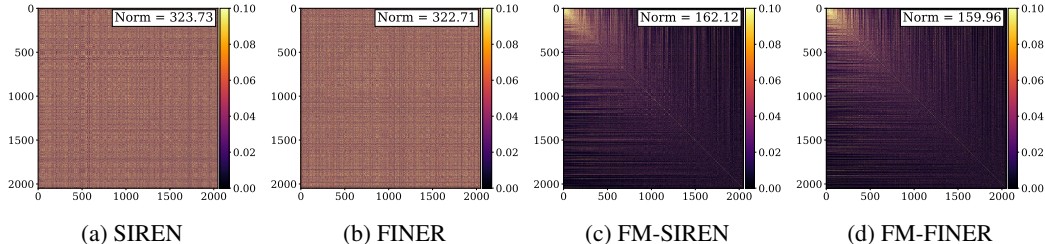

|          |          |          |          |
|----------|----------|----------|----------|
| (a) SIREN | (b) FINER | (c) FM-SIREN | (d) FM-FINER |

Figure 2: Covariance maps of hidden embeddings for different methods, with the Frobenius norm shown in the top-right of each subfigure. x and y axes represent neuron index. FM-SIREN (c) and FM-FINER (d) yield substantially lower covariance norms than SIREN (a) and FINER (b), reflecting improved frequency diversity. The stronger correlation observed in the upper-left corner of (c) and (d) corresponds to neurons associated with lower frequencies. All maps are derived from the networks used to reconstruct images in Figure 1.

## 4 PROPOSED METHOD

INR activations, such as sinusoids, frequency-variable sinusoids, and wavelets, offer substantial performance gains over classical activations like ReLU(.), sigmoid(.), and tanh(.) (Parhi & Nowak, 2020). However, MLPs employing these periodic activations still suffer from low frequency bias (Ramasinghe et al., 2022). This bias leads to redundancy between hidden embeddings, reducing the effective capacity of MLP linear layers, as illustrated in Figure 1. In this section, we propose a simple, yet effective, frequency multiplier scheme to address this problem, and show the resultant feature diversity.

### 4.1 NYQUIST-INFORMED FREQUENCY MULTIPLIER

The classical periodic bases (sines or cosines) employed in signal reconstruction (Equation 1) are predefined to span a fixed frequency range. According to the Nyquist theorem, this frequency span is determined by the sampling rate as:

$$f_{\text{span}} = (-f_{\text{Nyquist}}, +f_{\text{Nyquist}}) \tag{5}$$

We propose assigning a distinct frequency multiplier to each activation function, sampled from the frequency span defined by the Nyquist theorem. Since most digital signals, such as audio, 2D images, and 3D volumes, are real-valued, their spectra are symmetric around the zero frequency (Deppisch & Ahrens, 2025). Consequently, we restrict the multiplier range to $(0, +f_{\text{Nyquist}})$, as negative frequency components are redundant. Formally, the frequency multiplier and activation function for the $k^{th}$ neuron are defined as:

$$\phi_k(x) = \sin(\omega_k x), \qquad \omega_k = \frac{k f_{\text{Nyquist}}}{K}, \quad k = 0, 1, \ldots, K - 1 \tag{6}$$

This frequency multiplier is not a tunable hyperparameter, as it is derived directly from the Nyquist theorem. It can be applied to any periodic activation function. Moreover, our method is simple to implement and produces improved results over the baselines, as shown in the next section. We also propose to factor each frequency multiplier by $2/3$ for FM-FINER due to the wider frequency spectrum of FINER, reducing aliasing and improving performance, as shown in the ablation study.

### 4.2 FEATURE REDUNDANCY

Current periodic activation INRs, such as SIREN and FINER, apply the same activation function to all neurons within a layer. This uniformity limits the expressive capacity of the layer, in part due to the near-frequency bias of neural networks (Ramasinghe et al., 2022). Figures 2a and 2b show the covariance matrices of hidden features from networks used to reconstruct the images in Figure 1. The covariance values reveal substantial feature redundancy within a single layer. Ideally, the covariance matrix norm should approach zero, indicating orthogonality among neuron embeddings, but this is rarely achieved in practice. In contrast, our proposed networks, FM-SIREN and FM-FINER,

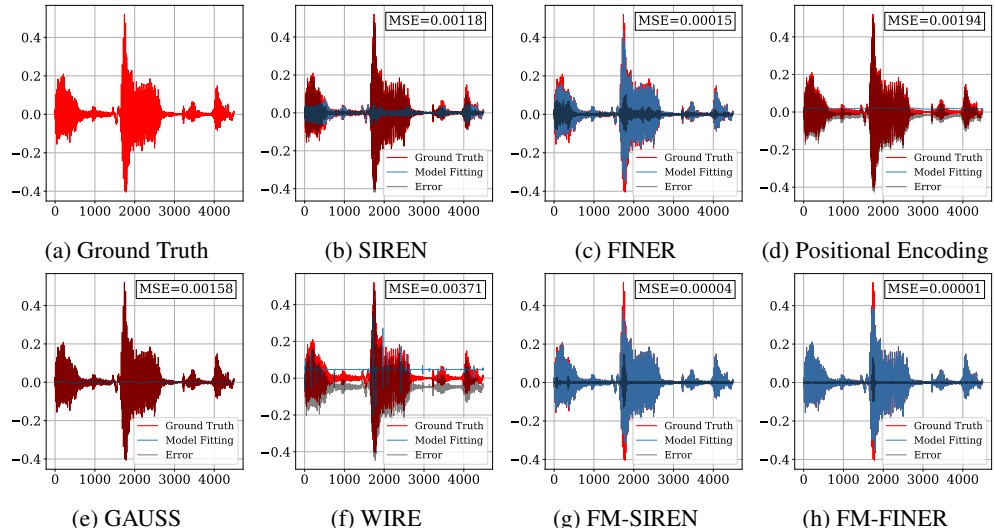

Figure 3: Qualitative results of one-second audio reconstructions for the 1995 Pacific Grand Prix clip in the Spoken English Wikipedia dataset (Köhn et al., 2016), using two-layer networks with different approaches. The one-second MSE of each reconstruction is reported in the top-right corner of its subfigure. FM-SIREN and FM-FINER achieve visibly closer alignment to the ground truth and substantially lower MSE compared to the baselines. Red , medium blue , black lines correspond to ground truth, reconstructed signal, and error signal, respectively.

Table 1: Average MSE for audio fitting on the Spoken English Wikipedia dataset (Köhn et al., 2016), comparing different INR models. Best and Second Best results are highlighted.

| Model | FINER | GAUSS | Positional Encoding | SIREN | WIRE | FM-SIREN | FM-FINER |
|---|---|---|---|---|---|---|---|
| **MSE** ↓ | $2.631 \times 10^{-4}$ | $5.948 \times 10^{-3}$ | $6.884 \times 10^{-3}$ | $5.138 \times 10^{-4}$ | $5.812 \times 10^{-3}$ | $4.738 \times 10^{-5}$ | $4.055 \times 10^{-5}$ |
| **Size** | $66,561$ | $66,561$ | $197,377$ | $66,561$ | $66,561$ | $66,561$ | $66,561$ |
| **Train Time (s)** | $0.61$ | $0.87$ | $0.84$ | $0.54$ | $2.31$ | $0.54$ | $0.61$ |

substantially reduce feature redundancy, as shown in Figures 2c and 2d, achieving a $49.92\%$ and $50.43\%$ improvement in the Frobenius norm of the covariance matrix compared to their baselines. Specifically, we generate a distinct feature vector of length 10,000 from each neuron, computing the covariance matrix of those feature vectors, and computing the Frobenius norm (Böttcher & Wenzel, 2008) of the resultant covariance matrix.

## 5 EXPERIMENTAL RESULTS

We evaluate the proposed FM-SIREN and FM-FINER across four representation tasks: 1D audio, 2D image, 3D shape, and NeRF. For fair comparisons, we adopt the hyperparameters recommended in each of the original works of the respective baselines. An exception is the audio task, where prior studies have not consistently benchmarked audio fitting; in this case, we selected the best-performing settings. To ensure efficiency and comparability, training is capped at 500 epochs for audio and image tasks, 75 epochs for 3D volumes, and 16 epochs for NeRF scenes. Full details of the training infrastructure, configurations, complete hyperparameter lists, ablation study, and additional results are provided in the appendix.

### 5.1 AUDIO FITTING

We evaluate audio reconstruction using two-layer models with 256 neurons in each layer. We used the Spoken English Wikipedia dataset Köhn et al. (2016) as a reference, which contains 1,313 clips sampled at $f_s = 4\,\text{kHz}$, informing a Nyquist frequency of $f_{\text{Nyquist}} = 2\,\text{kHz}$,. For each clip, we fit the first 10 seconds of audio and report performance in terms of mean squared error (MSE), averaged across all clips. As summarized in Table 1, FM-SIREN and FM-FINER achieve substantially lower errors than all baselines, and FM-FINER achieves the best overall performance. This demonstrates the effectiveness of Nyquist-informed frequency diversity in capturing fine-grained temporal struc-

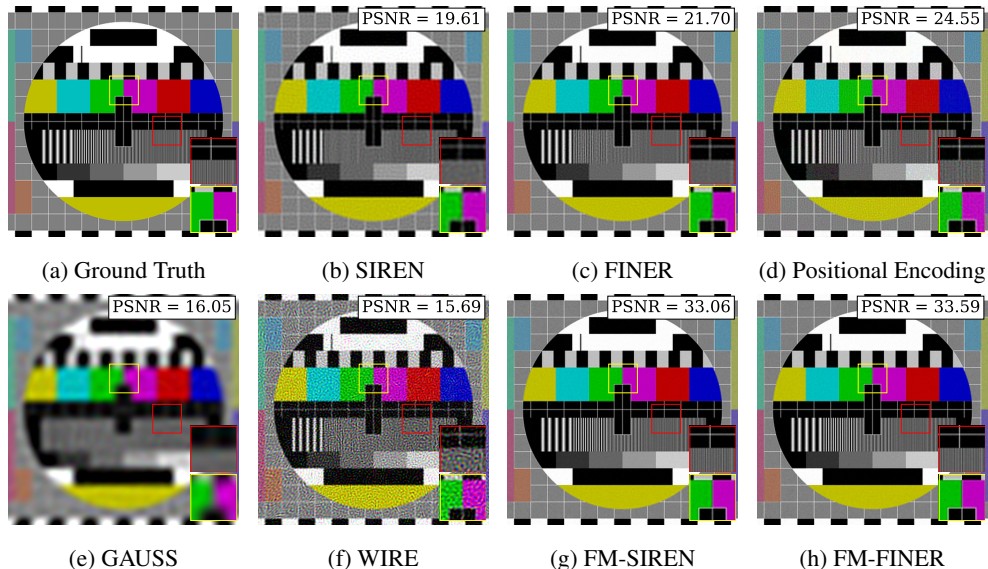

Figure 4: Qualitative image reconstruction results of the Philips Circle Pattern (Wikipedia contributors, 2025) using two-layer networks. The PSNR (dB) of each reconstruction is reported in the top-right corner of its subfigure. FM-SIREN and FM-FINER achieve the highest PSNR values and produce visibly sharper reconstructions compared to the baselines. While positional encoding attains the best performance among the baseline methods, its reconstructions appear noisier than ours, as highlighted in the zoomed-in slices.

Table 2: Average reconstruction performance on the Kodak (Mehta, 2020) and BSDS500 (Martin et al., 2001) datasets in MSE, PSNR, and SSIM. Best and Second Best results are highlighted.

| Model | Train Time (s) | Parameters | MSE $(10^{-3})$ ↓ | | PSNR (dB) ↑ | | SSIM ↑ | |
|---|---|---|---|---|---|---|---|---|
| | | | Kodak | BSDS500 | Kodak | BSDS500 | Kodak | BSDS500 |
| FINER | 1.12 | 67,331 | 1.808 | 0.881 | 28.236 | 31.468 | 0.778 | 0.887 |
| GAUSS | 1.18 | 67,331 | 2.690 | 1.552 | 26.472 | 29.024 | 0.695 | 0.817 |
| Positional Encoding | 1.28 | 197,891 | 3.140 | 2.706 | 25.810 | 26.884 | 0.705 | 0.799 |
| SIREN | 0.96 | 67,331 | 2.555 | 1.432 | 26.769 | 29.492 | 0.743 | 0.857 |
| WIRE | 3.27 | 67,331 | 1.071 | 0.433 | 29.874 | 33.921 | 0.763 | 0.888 |
| **FM-SIREN** | 0.96 | 67,331 | 0.660 | 0.175 | 32.285 | 38.115 | 0.874 | 0.967 |
| **FM-FINER** | 1.12 | 67,331 | 0.631 | 0.201 | 32.475 | 37.465 | 0.868 | 0.958 |

tures. Figure 3 provides qualitative results on a one-second segment of the 1995 Pacific Grand Prix clip, where our models exhibit a markedly lower reconstruction error compared to SIREN, FINER, and other baselines.

## 5.2 IMAGE FITTING

We evaluated all models on the Kodak Lossless True Color Image Suite (Mehta, 2020) and the BSDS500 dataset (Martin et al., 2001), which correspond to informed Nyquist frequencies of 256 and 160.5 cycles/image, respectively. Performance was evaluated using three standard metrics: MSE, peak signal-to-noise ratio (PSNR) (Korhonen & You, 2012), and structural similarity index (SSIM) (Wang et al., 2004). All models were implemented with two layers of 256 neurons each. As shown in Table 2, our FM-SIREN and FM-FINER consistently achieved the lowest MSE, highest PSNR, and highest SSIM, outperforming all baseline methods. In addition, we evaluated the models on the Philips Circle Pattern (Wikipedia contributors, 2025), a benchmark with sharp geometric structures ideal for testing reconstruction fidelity with a Nyquist frequency of 200 cycles/image. Figure 4 shows qualitative results, where FM-SIREN and FM-FINER exceed the strongest baseline by 8.51dB and 9.04dB in terms of PSNR, respectively. These results highlight the ability of our models to deliver high-fidelity image reconstructions using only two-layer networks. Additional reconstruction results along with their error distributions are provided in the appendix.

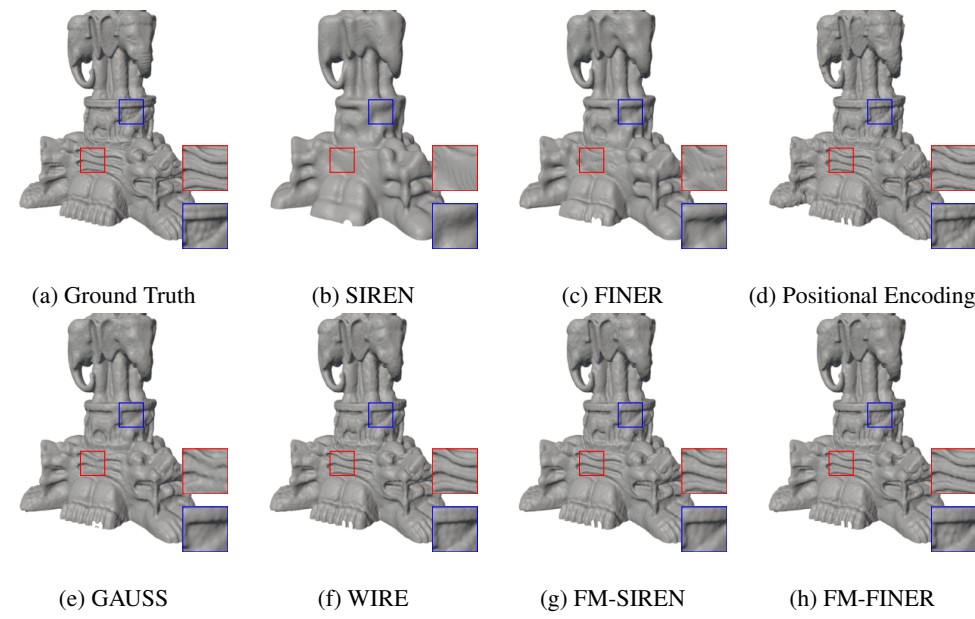

(a) Ground Truth  (b) SIREN  (c) FINER  (d) Positional Encoding

(e) GAUSS  (f) WIRE  (g) FM-SIREN  (h) FM-FINER

Figure 5: Qualitative reconstruction results for the Thai Statue 3D scene using three-layer networks. FM-SIREN and FM-FINER deliver higher-fidelity reconstructions with visibly sharper details compared to baselines. The zoomed-in slices highlight that fine structures are better preserved by our models, whereas WIRE, the third-best method, fails to capture fine detail in the blue zoomed-in slice.

Table 3: Epoch time, number of parameters, and IoU results for shape fitting on scenes from the Stanford 3D Scanning Repository dataset (Stanford Computer Graphics Laboratory, 2014). Best and Second Best results are highlighted.

| Model | Epoch Time (sec) | Parameters | IoU ↑ | | | |
|---|---|---|---|---|---|---|
| | | | Thai Statue | Armadillo | Dragon | Asian Dragon |
| FINER | 2.30 | 132, 865 | 0.976 | 0.991 | 0.994 | 0.978 |
| GAUSS | 3.14 | 132, 865 | 0.981 | 0.992 | 0.955 | 0.981 |
| Positional Encoding | 2.14 | 263, 169 | 0.982 | 0.994 | 0.977 | 0.980 |
| SIREN | 2.11 | 132, 865 | 0.960 | 0.980 | 0.981 | 0.956 |
| WIRE | 4.10 | 132, 865 | 0.988 | 0.994 | 0.993 | 0.974 |
| **FM-SIREN** | 2.11 | 132, 865 | 0.990 | 0.996 | 0.995 | 0.987 |
| **FM-FINER** | 2.30 | 132, 865 | 0.991 | 0.997 | 0.996 | 0.991 |

## 5.3 3D Shape Fitting

For 3D shape fitting, we follow the framework of Saragadam et al. (2023). Specifically, 3D shapes are sampled over a $512 \times 512 \times 512$ grid, where voxels inside the shape volume are assigned a value of $1$ and those outside are assigned $0$. This resolution corresponds to a Nyquist frequency of 256 cycles/volume. Reconstruction performance is evaluated using the Intersection over Union (IoU) metric (Rezatofighi et al., 2019). All models consist of three layers with 256 neurons each. Table 3 reports IoU scores for four shapes from the Stanford 3D Scanning Repository dataset (Stanford Computer Graphics Laboratory, 2014), where our proposed models consistently outperform the baselines. Figure 5 shows qualitative results for the Thai Statue, where FM-SIREN and FM-FINER yield visibly superior reconstruction fidelity, especially in the zoomed-in regions. Notably, Table 3 and Figure 5 also show that FM-SIREN and FM-FINER achieve these improvements without increasing compute time or parameter count.

## 5.4 Neural Radiance Fields

NeRF synthesizes novel 3D scene views from sparse 2D images by learning a continuous volumetric representation optimized via ray-based rendering (Mildenhall et al., 2020). In our experiments, the

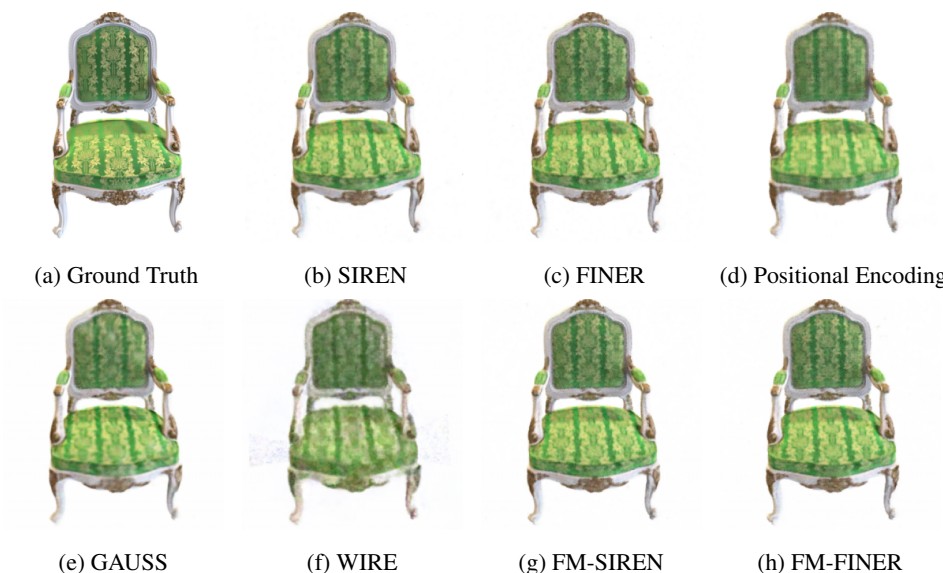

(a) Ground Truth      (b) SIREN      (c) FINER      (d) Positional Encoding

(e) GAUSS      (f) WIRE      (g) FM-SIREN      (h) FM-FINER

Figure 6: Qualitative reconstruction results of the Chair scene from the Blender dataset (Mildenhall et al., 2021), using five-layer networks (three layers for the first block, one layer for density, and one layer for color). FM-SIREN and FM-FINER achieve improvements over the baselines, with exact PSNR values reported in Table 4.

volume is discretized at a resolution of $100 \times 100 \times 100$, corresponding to a Nyquist frequency of 50 cycles/volume. We adopt the concise implementation of Vandegar (2023) on the Blender dataset (Mildenhall et al., 2021), using networks with five layers that model density and color. Performance is evaluated as the average PSNR across all 200 test images per scene, each at a resolution of $400 \times 400$. As reported in Table 4 and illustrated in Figure 6, FM-SIREN and FM-FINER achieve higher PSNR than baseline models in most scenes. Although the improvements in reconstruction quality are modest, both FM-SIREN and FM-FINER reduce compute time by $14.53\%$ and $17.06\%$, respectively, demonstrating the efficiency gains of our approach.

Table 4: Epoch time, number of parameters, and PSNR for NeRF fitting on the Blender dataset (Mildenhall et al., 2021). Best and Second Best results are highlighted.

| Model | Epoch Time (s) | Parameters | PSNR ↑ | | | | | | | |
|---|---|---|---|---|---|---|---|---|---|---|
| | | | Lego | Ship | Chair | Mic | Materials | Hotdog | Ficus | Drums |
| FINER | 106.84 | 199,428 | 28.01 | 33.09 | 34.53 | 34.44 | 33.38 | 37.16 | 33.75 | 32.51 |
| GAUSS | 98.68 | 199,428 | 27.58 | 32.91 | 34.23 | 34.48 | 33.21 | 36.66 | 33.33 | 32.31 |
| Positional Encoding | 126.33 | 214,788 | 27.63 | 32.48 | 34.13 | 34.24 | 32.86 | 36.17 | 32.82 | 31.72 |
| SIREN | 122.12 | 199,428 | 27.81 | 33.08 | 34.42 | 34.45 | 33.38 | 37.00 | 33.79 | 32.48 |
| WIRE | 140.04 | 199,428 | 23.36 | 20.34 | 25.21 | 26.31 | 23.07 | 27.50 | 23.19 | 21.11 |
| **FM-SIREN** | 104.37 | 199,428 | 27.55 | 33.17 | 34.55 | 34.64 | 33.45 | 37.37 | 33.76 | 32.54 |
| **FM-FINER** | 88.61 | 199,428 | 28.11 | 33.14 | 34.56 | 29.81 | 33.41 | 37.41 | 33.73 | 32.53 |

## 6 CONCLUSION

In this work, we tackled the issue of hidden feature redundancy in SOTA INR models such as SIREN and FINER. Our analysis showed that using the same frequency multiplier across neurons leads to correlated embeddings and limited capacity. To overcome this, we introduced FM-SIREN and FM-FINER, which assign Nyquist-informed, neuron-specific frequency multipliers. This simple modification reduces redundancy by nearly $50\%$ and yields consistent improvements across 1D audio, 2D image regression, 3D shape fitting, and NeRF synthesis. Beyond higher accuracy, our results highlight that frequency diversity within a single layer can significantly improve representational power without increasing network size or training time. In addition, we observed reduction in training time in NeRF fitting while improving performance. We believe this work facilitates new opportunities for bridging classical signal processing and modern implicit neural representations, paving the way for more efficient and scalable models.

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

## A  EXPERIMENTAL SETUP

### A.0.1  EXPERIMENTATION INFRASTRUCTURE

For all experiments, we used PyTorch Paszke et al. (2019) with Adam optimizer Kingma & Ba (2015), and a PyTorch StepLR scheduler which decayed the learning rate by a factor of 0.1 every 100 epochs. The audio and image experiments were performed on a Nvidia GTX 1080 Ti graphical processing unit (GPU) with 11GB memory and 32GB of system memory. 3D shape fitting and neural radiance fields experiments were performed on a Nvidia H200 GPU with 80GB memory and 256GB of system memory.

### A.0.2  EVALUATION METRICS

We used mean square error (MSE) to evaluate the predictive performance, which is expressed as:

$$\text{MSE} = \frac{\sum_{i=1}^{N}\left(x_i - \hat{x}_i\right)^2}{N} \tag{7}$$

where $x_i$ is the actual observation, $\hat{x}_i$ is the predicted value, and $N$ is the number of test points.

For images, we also used the Peak Signal-to-Noise Ratio (PSNR) and the Structural Similarity Index (SSIM). PSNR is a common metric to measure reconstruction quality. It is expressed in decibels (dB), and a higher value indicates a better fit. PSNR is defined as:

$$PSNR = 10 \cdot \log_{10}\left(\frac{MAX_I^2}{MSE}\right) \tag{8}$$

where $MAX_I$ is the maximum possible pixel value of the image (e.g., 255 for 8-bit images).

SSIM is a crucial metric that evaluates the similarity between two images on the basis of luminance, contrast, and structure. The index ranges from -1 to 1, with 1 indicating perfect similarity. SSIM is defined as:

$$SSIM(x, y) = \frac{(2\mu_x\mu_y + c_1)(2\sigma_{xy} + c_2)}{(\mu_x^2 + \mu_y^2 + c_1)(\sigma_x^2 + \sigma_y^2 + c_2)} \tag{9}$$

where $\mu_x$ and $\mu_y$ are the average pixel values, $\sigma_x$ and $\sigma_y$ are the standard deviations, $\sigma_{xy}$ is the covariance of the images $x$ and $y$, and $c_1$, $c_2$ are small constants to prevent division by zero.

For 3D shape fitting, we report the Intersection over Union (IoU), which measures the overlap between the predicted shape volume $V_p$ and the ground-truth volume $V_{gt}$. It is defined as:

$$IoU = \frac{|V_p \cap V_{gt}|}{|V_p \cup V_{gt}|} \tag{10}$$

where $\cap$ denotes the intersection and $\cup$ the union of occupied voxels. A higher IoU indicates better reconstruction fidelity of the 3D shape.

## B  ABLATION STUDY

We conducted extensive ablation experiments to assess the impact of three key factors: network width, network depth, and the Nyquist factor (i.e., the maximum frequency multiplier relative to the Nyquist frequency). The results for each factor are presented in the following subsections.

### B.1  NETWORK WIDTH

We conducted experiments with different numbers of neurons, $\{128, 256, 512, 1024\}$, in each layer for both our models and the baselines, while fixing the depth to two layers. Figure 7a illustrates the image fitting performance across all configurations. FM-SIREN and FM-FINER consistently

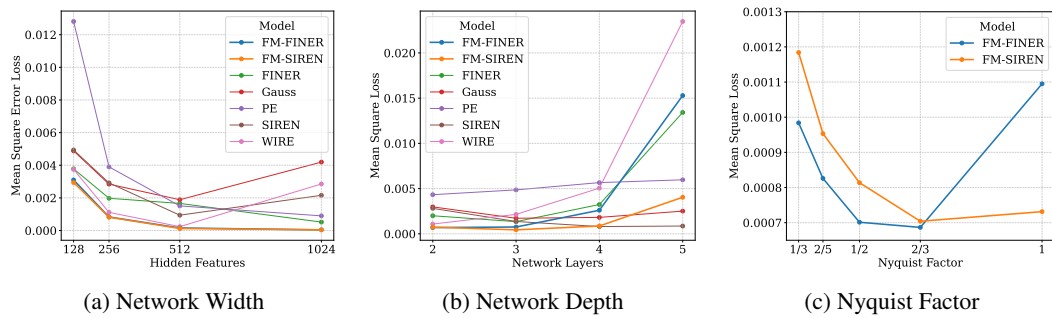

|                    |                   |                    |
|:------------------:|:-----------------:|:------------------:|
| (a) Network Width  | (b) Network Depth  | (c) Nyquist Factor |

Figure 7: Ablation study for different factors in the network design.

outperform the baselines at every width setting. However, their performance curves stagnate beyond 512 neurons. We attribute this to increased redundancy among hidden features, as the neuron-specific frequency multipliers are capped by the Nyquist range, leading to overlapping frequency components at larger widths.

## B.2 NETWORK DEPTH

We evaluated the effect of network depth for the number of layers, in the set $\{2, 3, 4, 5\}$, while fixing the network width to 256 neurons. Figure 7b reports the performance of all models for each depth. For FM-SIREN, the performance curve exhibits an inverse relationship with depth after three layers. We attribute this to the fact that FM-SIREN introduces higher-frequency components in the early layers, which propagate as noise in deeper layers and reduce overall accuracy. FM-FINER shows a similar trend but with a steeper decline, reflecting its greater sensitivity to depth. Notably, despite this degradation at larger depths, both FM-SIREN and FM-FINER outperform the best baseline even with significantly shallower architectures. For example, a two-layer FM-SIREN or FM-FINER matches or outperforms a five-layer SIREN.

## B.3 NYQUIST FACTOR

The Nyquist theorem defines the maximum frequency that can be represented in a digital signal without aliasing. To study its effect, we tested different ranges of frequency multipliers relative to the Nyquist frequency, $\{1/3, 2/5, 1/2, 2/3, 1\}$. Figure 7c presents the performance curves of FM-SIREN and FM-FINER across these factors. For FM-SIREN, performance generally improves proportionally with the Nyquist factor. In contrast, FM-FINER shows improvements only up to a factor of $2/3$, beyond which performance saturates. We attribute this to its chirp-based activations, which inherently possess a broader frequency range (bandwidth) than sinusoidal activations that causes aliasing with large multipliers. Overall, both models validate our central hypothesis: explicitly incorporating Nyquist-informed frequency limits into activation functions enhances representational capacity and reduces redundancy.

Taken together with the width and depth studies, these results highlight that frequency diversity is the key driver of improved performance in FM-SIREN and FM-FINER.

## C HYPERPARAMETERS

All experiments share the same model-specific hyperparameters recommended in each baseline's reference with an exception for $\omega_0$. We used the best performing $\omega_0$ for audio fitting since not all references presented audio fitting experiments. Table 5 presents global, model-specific hyperparameters, whereas 6 presents experiment-specific hyperparameters.

Table 5: Model-specific hyperparameters for the all experiments.

| Parameter | SIREN | FINER | WIRE | Gauss | PE | FM-SIREN | FM-FINER |
|---|---|---|---|---|---|---|---|
| First $\omega_0$ | 30.0 | 30.0 | 16 | - | - | 30.0 | 30.0 |
| Hidden $\omega_0$ | 30.0 | 30.0 | 16 | - | - | 30.0 | 30.0 |
| Outermost Linear | ✓ | - | - | - | - | ✓ | - |
| Bias | ✓ | ✓ | ✓ | ✓ | - | ✓ | ✓ |
| Scale | - | - | 8 | 16 | 15 | - | - |
| Embedding Size | - | - | - | - | 256 | - | - |
| Nyquist Factor | - | - | - | - | - | 1 | 2/3 |

Table 6: Experiment-specific hyperparameters.

| Parameter | Audio | Image | 3D Shape | NeRF |
|---|---|---|---|---|
| FINER $\omega_0$ | 700 | 30 | 30 | 30 |
| SIREN $\omega_0$ | 800 | 30 | 30 | 30 |
| WIRE $\omega_0$ | 16 | 16 | 16 | 16 |
| FM-SIREN $\omega_0$ | 800 | 30 | 30 | 30 |
| FM-FINER $\omega_0$ | 700 | 30 | 30 | 30 |
| Learning Rate | 0.0001 | 0.001 | 0.001 | 0.0001 |
| Number of Hidden Layers | 2 | 2 | 3 | 5 |
| Number of Epochs | 500 | 500 | 75 | 16 |

# D ADDITIONAL RESULTS

We provide additional experimental results of the models discussed in the main paper. These results further demonstrate the effectiveness of our proposed frequency modulation approach in enhancing the fitting capabilities of implicit neural representation. Our quantitative audio fitting results in Table 7 show a superior performance in terms of MSE, which is further supported by the visual comparisons in Figures 8 to 11. For image fitting, we present supplementary quantitative results for both PSNR and SSIM in Table 8, and Figures 12 to 15 visually confirm the higher quality of reconstruction achieved by our models. Finally, qualitative results for 3D shape fitting and neural radiance fields demonstrate our models' ability to capture intricate details and produce higher-quality renderings as shown in Figures 16 to 19

Table 7: Sample MSE results for audio reconstruction from the Spoken English Wikipedia dataset (Köhn et al., 2016). Best , Second Best .

| Model | 1995 Pacific Grand Prix ($\times 10^{-3}$) | Alzheimer Disease ($\times 10^{-3}$) | Flag of Canada ($\times 10^{-3}$) | Functional Programming ($\times 10^{-3}$) | Munich ($\times 10^{-3}$) | Average ($\times 10^{-3}$) |
|---|---|---|---|---|---|---|
| FINER | 0.385 | 0.548 | 1.632 | 0.918 | 0.052 | 0.707 |
| GAUSS | 1.576 | 1.096 | 4.433 | 2.923 | 9.546 | 3.9148 |
| Positional Encoding | 1.981 | 1.575 | 4.914 | 3.269 | 9.968 | 4.3414 |
| SIREN | 1.458 | 0.989 | 4.395 | 2.748 | 0.932 | 2.1044 |
| WIRE | 1.895 | 1.089 | 5.052 | 2.910 | 11.108 | 4.4107 |
| **FM-SIREN** | 0.181 | 0.202 | 0.203 | 0.161 | 0.033 | 0.156 |
| **FM-FINER** | 0.014 | 0.023 | 0.016 | 0.009 | 0.001 | 0.0126 |

Table 8: Sample PSNR (dB) results for image reconstruction on images from Kodak and BSD500 datasets. Best , Second Best .

| Model | Kodak | | | BSD500 | | | Average |
|---|---|---|---|---|---|---|---|
| | kodim02 | kodim03 | kodim22 | 100075 | 113016 | 216041 | |
| FINER | 28.77 | 29.16 | 26.97 | 31.65 | 27.23 | 29.54 | 28.89 |
| GAUSS | 25.80 | 26.46 | 23.85 | 26.29 | 22.28 | 25.82 | 25.08 |
| Positional Encoding | 24.33 | 28.22 | 26.15 | 27.06 | 23.44 | 27.05 | 26.04 |
| SIREN | 27.21 | 27.38 | 24.95 | 28.37 | 24.05 | 27.14 | 26.52 |
| WIRE | 26.40 | 27.18 | 24.74 | 27.69 | 23.45 | 26.61 | 26.01 |
| **FM-SIREN** | 31.93 | 33.12 | 29.91 | 38.69 | 32.65 | 37.35 | 33.94 |
| **FM-FINER** | 32.59 | 32.90 | 30.23 | 38.45 | 32.11 | 36.33 | 33.77 |

## E  LLM USAGE

We did not use large language models (LLMs) for research ideation or for generating scientific content in this paper. LLMs were only used for light assistance in polishing the writing (e.g., improving grammar and readability).

## SUPPLEMENTARY REFERENCES

Diederik P. Kingma and Jimmy Ba. Adam: A method for stochastic optimization. In *International Conference on Learning Representations*, 2015. URL `https://arxiv.org/abs/1412.6980`.

Adam Paszke, Sam Gross, Francisco Massa, Adam Lerer, James Bradbury, Gregory Chanan, Trevor Killeen, Zeming Lin, Natalia Gimelshein, Luca Antiga, Alban Desmaison, Andreas Köpf, Edward Yang, Zachary DeVito, Martin Raison, Alykhan Tejani, Sasank Chilamkurthy, Benoit Steiner, Lu Fang, Junjie Bai, and Soumith Chintala. Pytorch: An imperative style, high-performance deep learning library. In H. Wallach, H. Larochelle, A. Beygelzimer, F. d'Alché Buc, E. Fox, and R. Garnett (eds.), *Advances in Neural Information Processing Systems*, volume 32, pp. 8024–8035. Curran Associates, Inc., 2019. URL `https://proceedings.neurips.cc/paper/2019/file/bdbca288fee7f92f2bfa9f7012727740-Paper.pdf`.

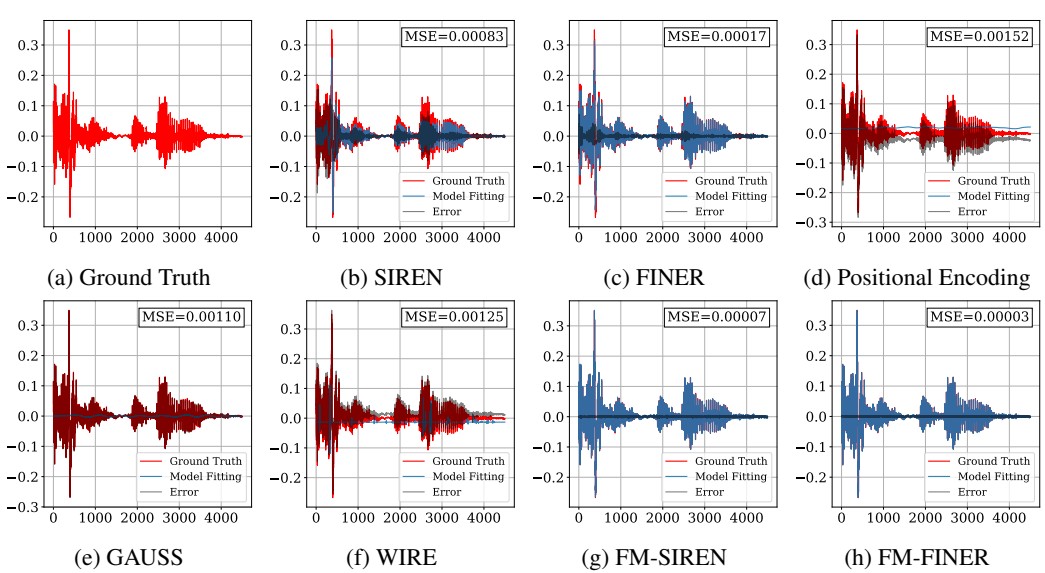

Figure 8: Qualitative results of one-second audio reconstructions for the Alzheimer27s Disease clip from the Spoken English Wikipedia dataset Köhn et al. (2016).

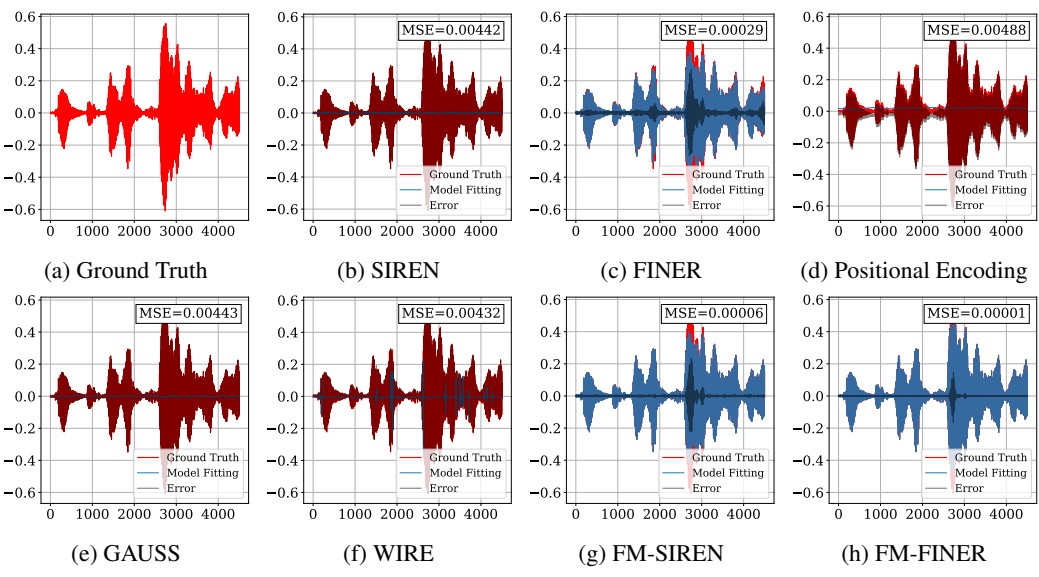

Figure 9: Qualitative results of one-second audio reconstructions for the Flag of Canada from the Spoken English Wikipedia dataset (Köhn et al., 2016).

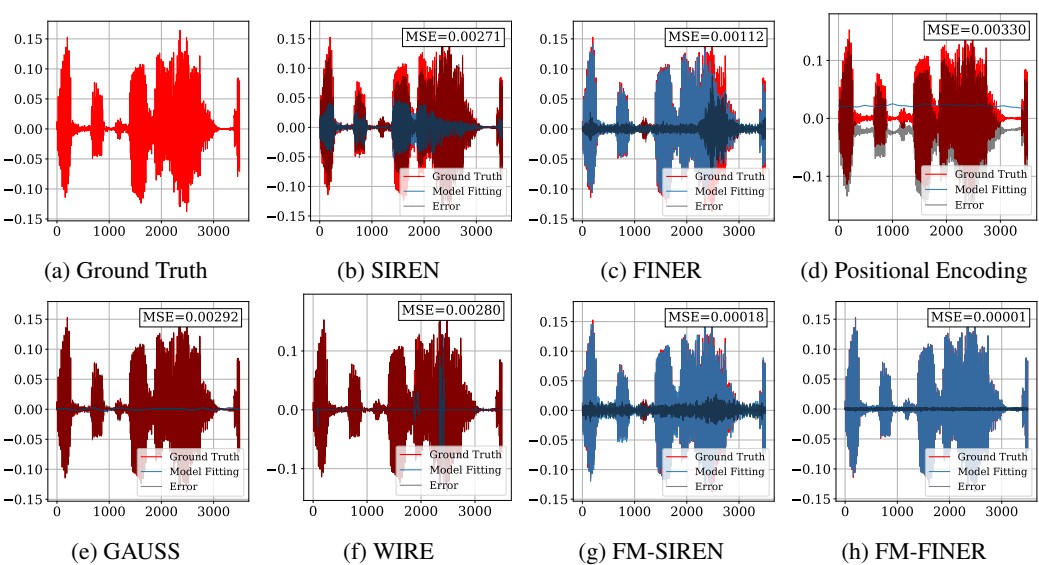

Figure 10: Qualitative results of one-second audio reconstructions for the Functional Programming from the Spoken English Wikipedia dataset (Köhn et al., 2016).

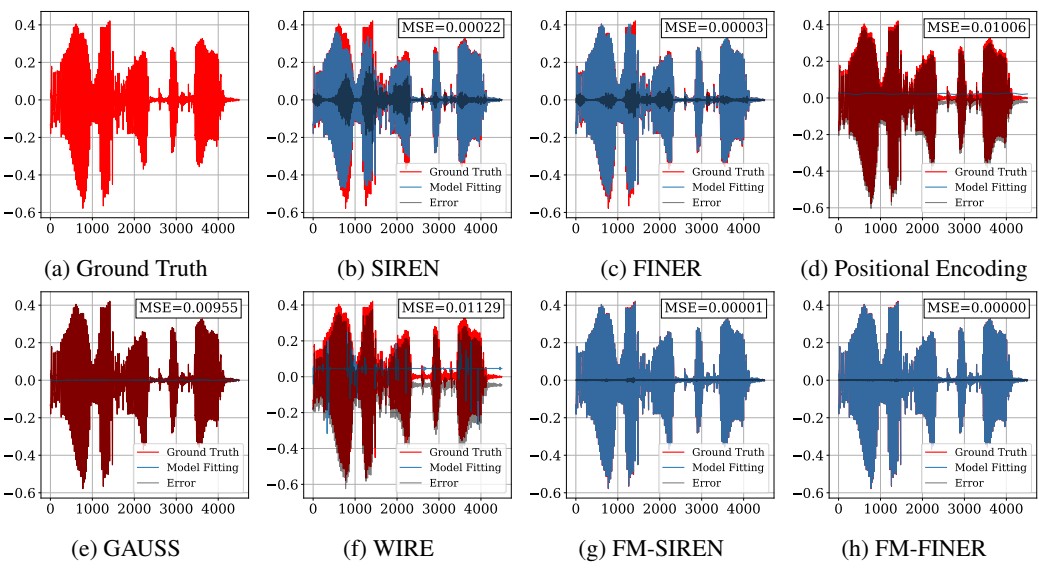

Figure 11: Fitting the a 10 second cut of the Munich from the Spoken English Wikipedia dataset (Köhn et al., 2016) on E-SIREN and E-FINER.

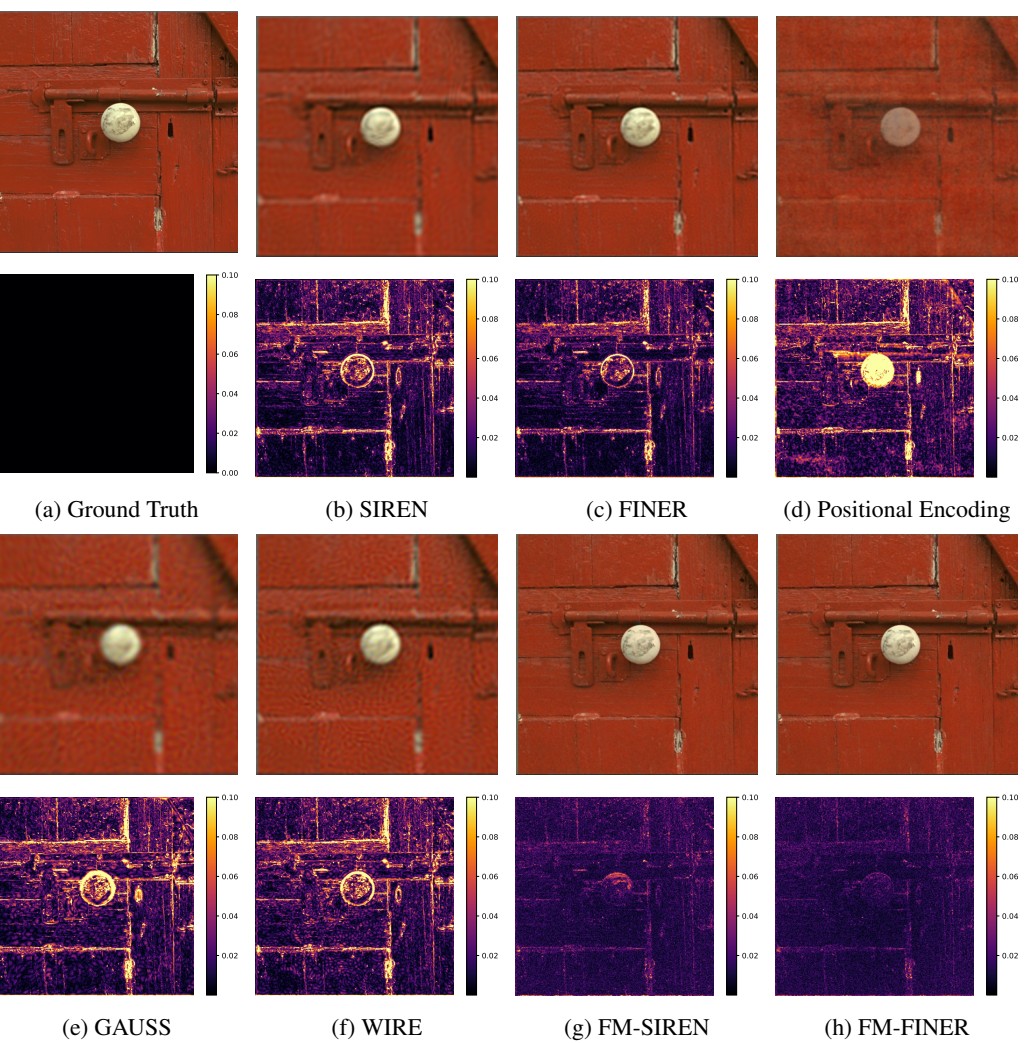

Figure 12: Qualitative results for kodim02 from Kodak Lossless True Color Image Suite (Mehta, 2020)). Top image in each subfigure is the reconstruction results while the bottom one is the error distribution between reconstruction and ground truth.

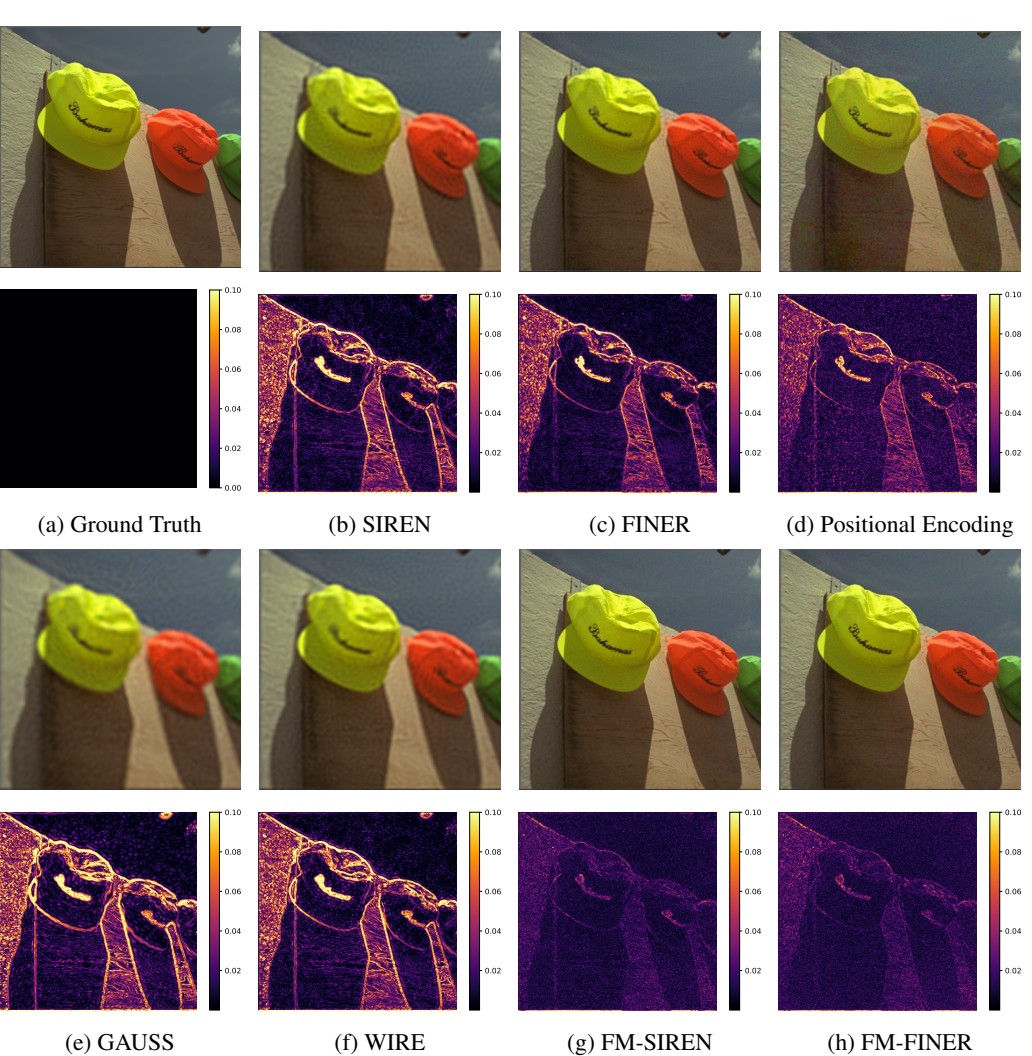

(a) Ground Truth  (b) SIREN  (c) FINER  (d) Positional Encoding

(e) GAUSS  (f) WIRE  (g) FM-SIREN  (h) FM-FINER

Figure 13: Qualitative results for kodim03 from Kodak Lossless True Color Image Suite (Mehta, 2020)). Top image in each subfigure is the reconstruction results while the bottom one is the error distribution between reconstruction and ground truth.

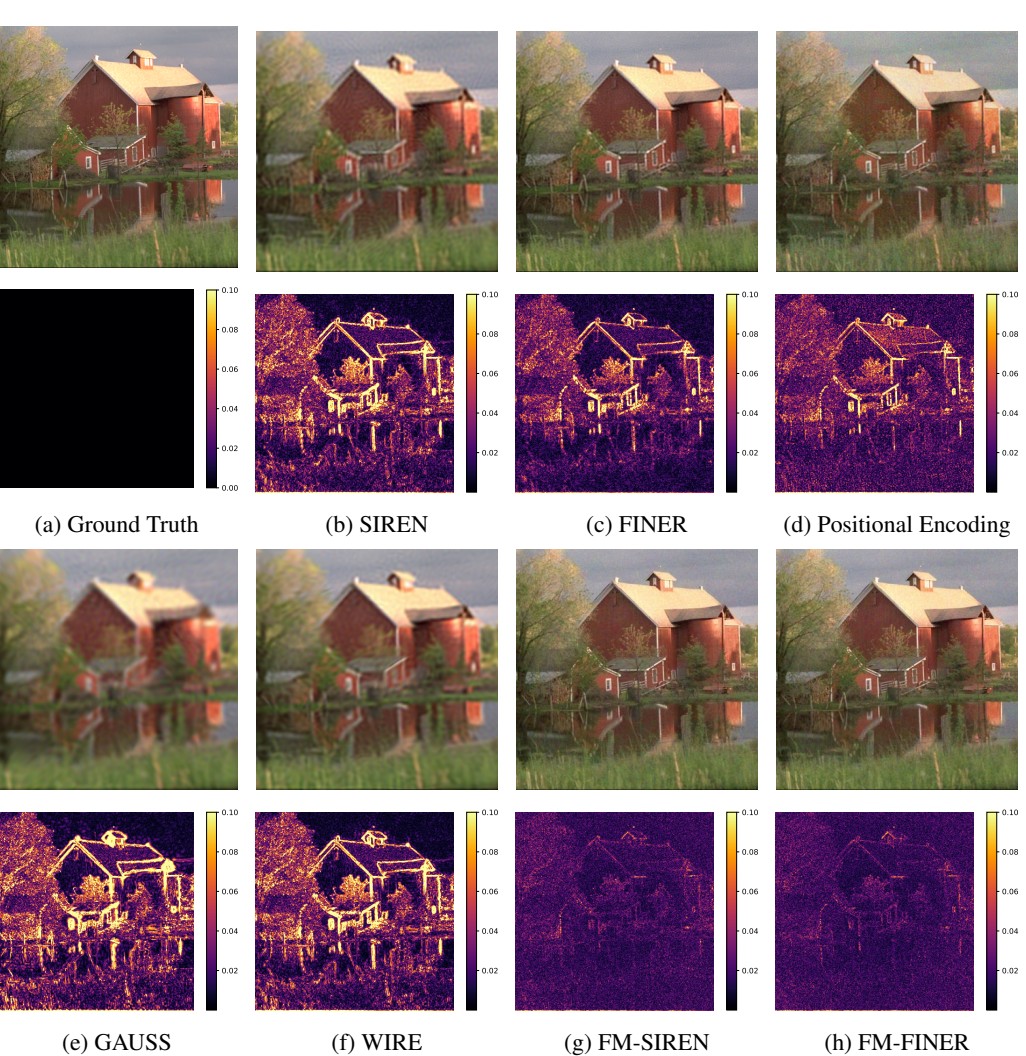

Figure 14: Qualitative results for kodim22 from Kodak Lossless True Color Image Suite (Mehta, 2020)). Top image in each subfigure is the reconstruction results while the bottom one is the error distribution between reconstruction and ground truth.

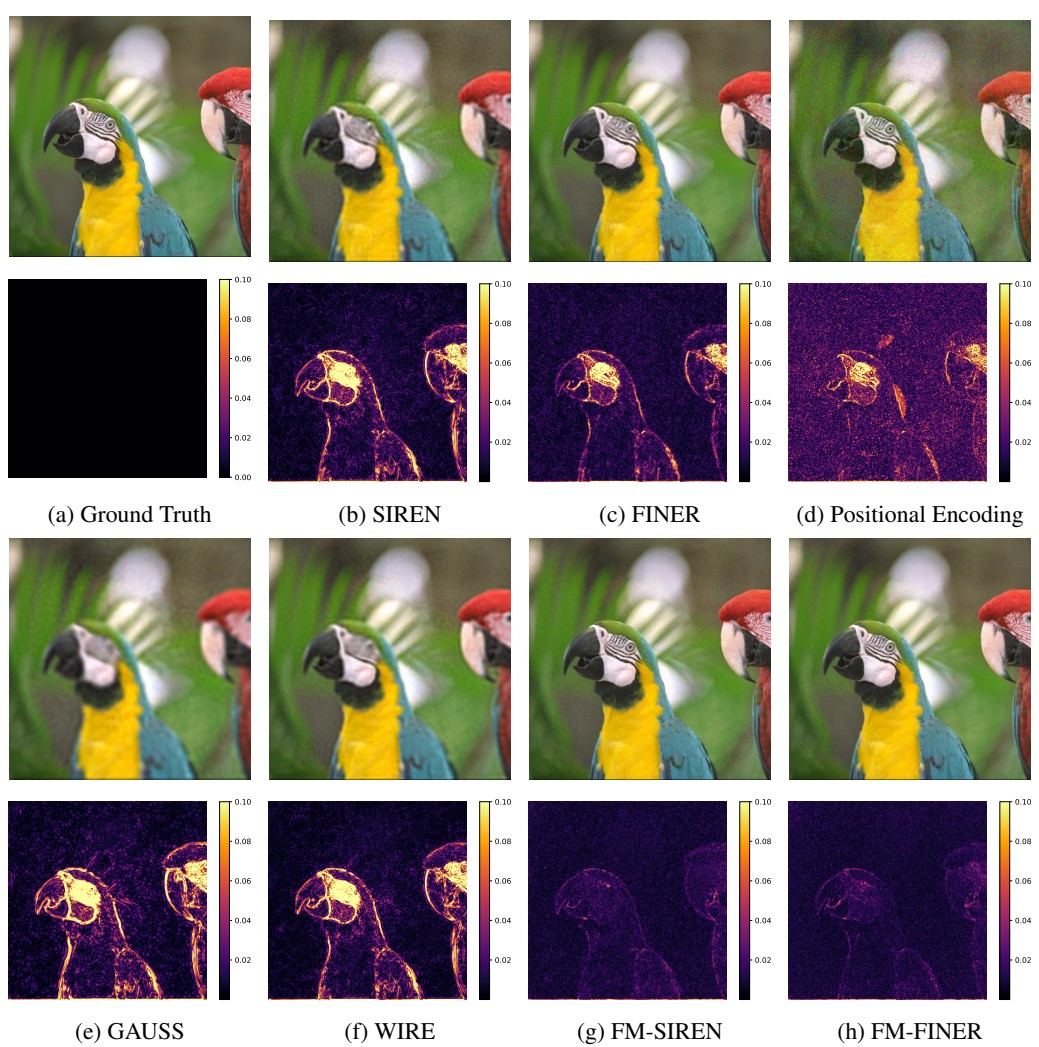

Figure 15: Qualitative results for kodim23 from Kodak Lossless True Color Image Suite (Mehta, 2020)). Top image in each subfigure is the reconstruction results while the bottom one is the error distribution between reconstruction and ground truth.

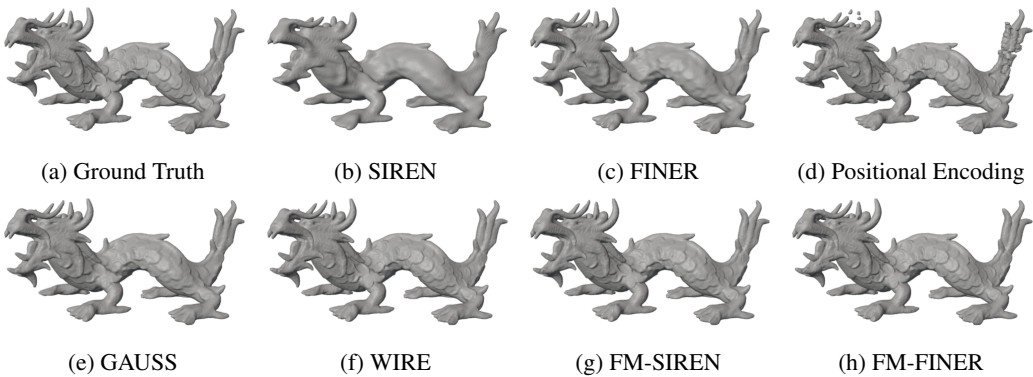

Figure 16: Qualitative 3D reconstruction results of the Asian Dragon scene from the Stanford 3D Scanning Repository dataset (Stanford Computer Graphics Laboratory, 2014).

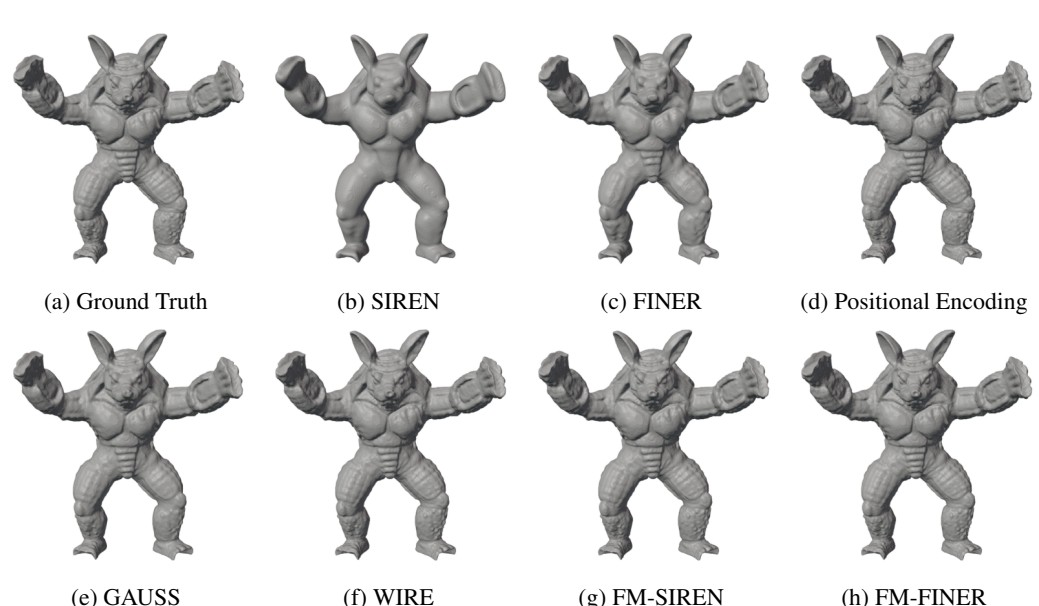

(a) Ground Truth      (b) SIREN      (c) FINER      (d) Positional Encoding

(e) GAUSS      (f) WIRE      (g) FM-SIREN      (h) FM-FINER

Figure 17: Qualitative 3D reconstruction results of the Armadillo scene from the Stanford 3D Scanning Repository dataset (Stanford Computer Graphics Laboratory, 2014).

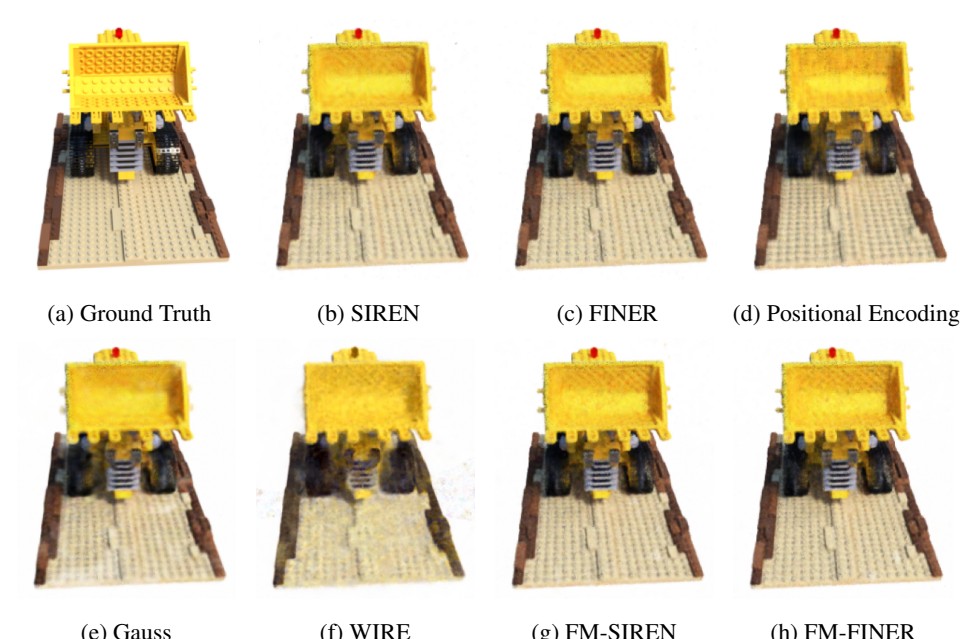

(a) Ground Truth      (b) SIREN      (c) FINER      (d) Positional Encoding

(e) Gauss      (f) WIRE      (g) FM-SIREN      (h) FM-FINER

Figure 18: Qualitative reconstruction results of the Lego scene from the Blender dataset (Mildenhall et al., 2021).

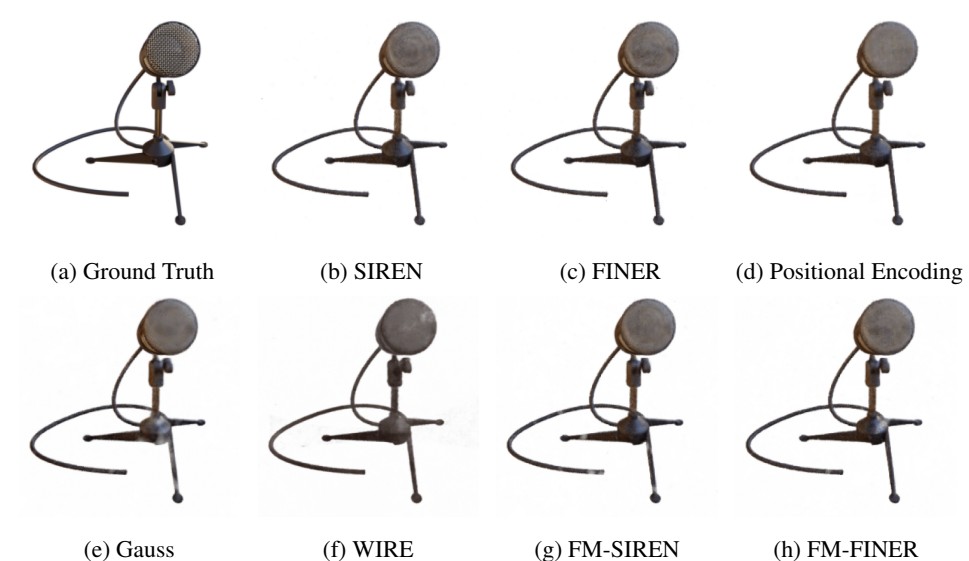

(a) Ground Truth        (b) SIREN        (c) FINER        (d) Positional Encoding

(e) Gauss        (f) WIRE        (g) FM-SIREN        (h) FM-FINER

Figure 19: Qualitative reconstruction results of the Mic scene from the Blender dataset (Mildenhall et al., 2021).

