# OpenReview forum: "FM-SIREN & FM-FINER: Nyquist-Informed Frequency Multiplier for Implicit Neural Representation with Periodic Activation"
_ICLR.cc/2026/Conference — Submitted to ICLR 2026_

### Official Review · Reviewer_JXTJ · 2025-10-21

**Soundness:** 2
**Presentation:** 2
**Contribution:** 2
**Rating:** 4
**Confidence:** 3

**Summary:**

Existing periodic activation-based implicit neural representations (INRs) like SIREN and FINER suffer from hidden feature redundancy, as all neurons in a layer share a fixed frequency multiplier. Drawing inspiration from classical signal processing, this paper introduces FM-SIREN and FM-FINER, which assign a unique, Nyquist-informed, neuron-specific frequency multiplier to each activation. This simple modification introduces frequency diversity without requiring hyperparameter tuning and reduces feature redundancy. As a result, the proposed models consistently improve signal reconstruction quality across diverse tasks, including 1D audio, 2D images, and 3D shapes.

**Strengths:**

1. Interesting Idea

The proposed solution, assigning neuron-specific frequencies based on the Nyquist theorem, is simple, elegant, and directly validated.

2. Strong Empirical Results

The method is evaluated across four different data modalities: 1D audio, 2D images, 3D shapes, and NeRF.

**Weaknesses:**

1. The general idea is not well formulated. The introduction of the model should be rewritten to focus on the problem and the proposed solution.
2. Figure 7b shows that the performance of both FM-SIREN and FM-FINER degrades when the network depth is increased beyond three layers.
3. Lack of competition with:
- SPDER: Semiperiodic Damping-Enabled Object Representation
- FreSh: Frequency Shifting for Accelerated Neural Representation Learning
4. The motivation of the paper and Figure 1 is weak. Authors should present results on more complicated images, with a larger network.
5. Authors do not use video data for evaluations.
6. The lack of a direct visualization of the proposed method.

**Questions:**

1. Why does the method's performance degrade in networks deeper than three layers?
2. How does assigning diverse frequencies compare to other methods, like FreSh, that find a single optimal frequency?
3. Why was the method not evaluated on video data, and how would the Nyquist multipliers be defined for video?
4. Can the authors provide a diagram visualizing how different frequencies are assigned to each neuron in a layer?
5. Will the implementation code be made publicly available to ensure reproducibility?

---

> ### Author Response · Authors · 2025-11-27
>
> We thank the reviewer for his/her valuable feedback. In the following, we address the reviewer’s concerns in the same order as Weaknesses and Questions.
>
> **[W1]** We appreciate this feedback. While our core idea is presented in Equation (6) and validated in Figure 2, we agree the introduction could better motivate the problem. As suggested by the reviewer, we will revise it to: (1) explicitly state that existing methods use fixed frequency multipliers that cause feature redundancy, (2) introduce our Nyquist-informed approach as a principled solution to single layer capacity.
>
> **[W2]** The degraded performance with deeper networks is insignificant when evaluating the practical value: FM-FINER and FM-SIREN with 2 layers outperforms 5-layer baseline SIREN and FINER (as shown in Figure 7b). This demonstrates our method enables 60% reduction in network depth while improving performance, which is a very significant practical advantage for deployment. Our method spans frequencies up to $f_{Nyquist}$ in the first layer. When these high-frequency components propagate through multiple layers, they create higher-order harmonics that exceed the Nyquist limit, causing aliasing and instability.
>
> **[W4]** Figure 1 demonstrates the lacking layer capacity in modern INR (SIREN and FINER), where this single linear layer, in principle, resembles classical signal reconstruction with periodic basis functions. The performance of shallow networks against this simple image reveals the of linear layer capacity in SIREN and FINER. Unlike SIREN and FINER, our models show significant performance improvement with shallow networks. Moreover, complicated INR models do not provide significant practical value such as compression. Therefore, we believe using good-performing shallow INR models is a practical motivation for the problem. Our contribution is demonstrating that with proper frequency diversity, shallow networks (2-3 layers) outperform deeper baselines. We believe this is a significant contribution in INR and and NeRF research. Figure 7b validates this design choice.
>
> **[W5]** We agree. We haven’t used video data in evaluation and we’d be happy to include it in the revised paper, as suggested by the reviewer.
>
> **[W6]** While we thought the method is composed of a simple, but informed, frequency multiplier to the activation functions which does not require complicated diagrams, we will include a diagram in our revised paper, as suggested by the reviewer.
>
> **[Q1]** The degradation occurs because each layer in our method spans a wider frequency range ($0$ to $f_{Nyquist}$) compared to baselines. When stacked deeply, high-frequency components from early layers propagate through subsequent layers, causing interference and noise accumulation, a known issue with periodic activations in deep networks. This validates our design principle: maximize frequency diversity within layers rather than relying on depth. Our 2-3 layer networks outperform deeper baselines by efficiently utilizing each layer's capacity.
> Mathematically, composing periodic functions f(g(x)) with wide frequency ranges creates high-order harmonics that exceed the Nyquist limit, causing aliasing.
>
> **[Q2]** Diverse frequencies increase the bandwidth of a single layer in the MLP by assigning each neuron a distinct frequency multiplier (ωₖ). FreSh depends solely on depth and optimization to widen its spectrum, whereas our approach explicitly guarantees frequency diversity within each layer through Nyquist-informed assignments. This enables shallower networks (2-3 layers) to achieve comparable or better performance than deeper architectures. Moreover, FreSh would suffer from feature redundancy since it uses single frequency as demonstrated in Figure 2.
>
> **[Q3]** The video data are just sequential images stored as a volume. This volume is discretized into a 3D grid, first and second dimensions are for the image size and the third is for the temporal sequence, which informs spectral limits according to Nyquist. For example, a video of 10 sequential images of resolution 256×256 constitutes a 3D volume of size 256×256×10. The frequency limits according to Nyquist are 128, 128, and 5. This informs a limited frequency multiplier up to 128.
>
> We did not include video experiments in the current submission as our 3D shape experiments (Table 3) already demonstrate the method's effectiveness on volumetric data. However, we agree with the reviewer that this would be a valuable addition and can include video fitting results in the revised paper, if helpful.
>
> **[Q4]** We can introduce a diagram of a simple MLP with multiplier present in it, as requested by the reviewer,
>
> **[Q5]** The implementation code will be publicly available upon publication.

---

### Official Review · Reviewer_zW1b · 2025-10-29

**Soundness:** 3
**Presentation:** 3
**Contribution:** 3
**Rating:** 4
**Confidence:** 5

**Summary:**

This paper introduces FM-SIREN and FM-FINER to address hidden feature redundancy. These networks improve frequency diversity by applying Nyquist-informed, neuron-specific frequency multipliers to periodic activations, all without requiring extra network depth or hyperparameter tuning. Experiments across 1D audio, 2D images, 3D volumes, and NeRF show consistent gains over SIREN/FINER, with ~50% reduction in hidden-feature redundancy.

**Strengths:**

1. The proposed method, the frequency multiplier, is simple yet effective for addressing the problem of hidden feature redundancy.

2. Experimental results across various INR tasks demonstrate consistent improvements over SIREN/FINER.

**Weaknesses:**

1. The epoch count seems too low for typical INR experiments, raising concerns that the network may not have fully converged.

2. The proposed Nyquist-informed, neuron-specific frequency multipliers may not be well-suited for low-frequency signals. For example, when fitting a very smooth $512\times512$ image, the resulting Nyquist frequency ($f_{Nyquist}=256$) could cause the network to introduce high-frequency artifacts.

3. In the Neural Radiance Field experiments, the improvement in reconstruction quality from FM-SIREN/FINER appears marginal. Furthermore, the experimental setup is confusing. For instance, the justification for discretizing the volume into a $100\times100\times100$ resolution is unclear. It is also not explained why FM-SIREN/FINER would reduce computation time compared to SIREN/FINER, given that, theoretically, the time per epoch should be identical.

4. The proposed method may be useful for signal representations where the Nyquist frequency is known a priori. However, for tasks where this prior is unknown, such as NeRF or solving PDEs, empirical hyperparameter tuning may still be necessary. How to choose a proper $f_{Nyquist}$?

**Questions:**

1. What are the performance metrics after the networks have fully converged?

2. Do FM-SIREN/FINER still provide performance gains when fitting a very smooth image?

3. Given that the hyperparameters ($\omega_k$) are determined by the Nyquist-informed, neuron-specific frequency multipliers, why is $\omega_0$ still listed as a hyperparameter in Appendix Table 6? Then, what exactly are the hyperparameter settings for FM-SIREN/FINER across the various tasks?

4. Are the proposed frequency multipliers applied only to the first layer, or to all hidden layers? This is not clearly specified in the paper.
See Weaknesses for other questions.

---

> ### Author Response · Authors · 2025-11-27
>
> We thank the reviewer for his/her feedback. In the following, we address the reviewer’s concerns in the same order as Weaknesses and Questions.
>
> **[W1]** All networks have fully converged. We will provide learning curves in the supplementary material demonstrating convergence for all baselines and our methods to address this concern of the reviewer. The consistent improvements across 500 epochs (audio/image) and 75 epochs (3D) indicate our gains are not due to convergence differences but genuine representational advantages from frequency diversity.
>
> **[W2]** Our proposed frequency multiplier spans the defined frequency spectrum of a signal, from 0 to the Nyquist frequency. The test images in our evaluation already include images with both smooth regions and high-frequency components, and our method shows superior qualitative performance compared to baselines.
>
> Specifically:
>
> • Figure 4 (Philips Circle Pattern): Contains both smooth circular regions and sharp high-frequency geometric patterns. Our method achieves 8.51dB (FM-SIREN) and 9.04dB (FM-FINER) improvement over the best baseline which is very significant, demonstrating excellent handling of both smooth and sharp features.
>
> • Figure 15 (kodim23): A natural image composed of smooth regions (parrots' bodies, blurred background) and high-frequency details (feathers, textures). The error distribution maps show our methods capture both smooth gradients and fine details more accurately than baselines.
>
> If the reviewer believes additional synthetic smooth-only images would strengthen the evaluation, we’d be happy to provide them, but our current natural images already demonstrate superior performance on smooth content as part of their composition.
>
> **[W3]** The product of the NeRF framework is a neural network that represents 3D volume of an object in terms of color and opacity. Since this 3D volume is discretized during training (like any other input images and 3D shapes), it has a predefined maximum frequency according to Nyquist theorem. Therefore, we apply the theorem based on the volume size. In our process, we adopt the standard 100×100×100 volume size commonly used in NeRF.
>
> Considering that NeRF learning is dependent on the learned ray density in each epoch, the lower training time shows that our FM-FINER and FM-SIREN learn the ray density more quickly than the baselines. Therefore, when density is high in earlier samples along the ray, farther samples can be skipped, as they are occluded behind the surface. This is known as early ray termination optimization.
>
> **[W4]** Solving PDEs includes defining a reconstruction grid, which itself has Nyquist limits. Therefore, $f_{Nyquist}$ is determined by the target grid resolution (e.g., for a 100×100 grid, $f_{Nyquist}$ = 50 per dimension), giving the target resolution of the grid we are fitting via the frequency multiplier. This is not a hyperparameter to tune but rather a property of the discretized problem—analogous to how image resolution directly determines f_Nyquist for image fitting.
>
> **[Q1]** Yes, all networks have converged. Our training schedules (500 epochs for audio/image, 75 for 3D, 16 for NeRF) ensure full convergence. The consistent improvements across all tasks demonstrate genuine representational advantages, not convergence artifacts. Learning curves can be provided in the revised version, if helpful.
>
> **[Q2]** Regarding smooth images: our evaluated datasets already contain substantial smooth regions (see W2 response - Figures 4 and 15). We can provide additional synthetic smooth images if the reviewer believes this would strengthen the evaluation.
>
> **[Q3]** ω₀ is used for initialization. As stated in the method section, our method uses the same initialization scheme as the original SIREN and FINER, which requires ω₀ to properly scale the initial weights. However, in the forward pass (activation functions), our Nyquist-informed frequency multipliers replace the role of ω₀.
>
> **[Q4]** All hidden layers. Each layer applies Equation 6 to assign neuron-specific frequency multipliers. We will clarify this in the revised paper, as suggested by the reviewer.

---

### Official Review · Reviewer_6yc9 · 2025-10-31

**Soundness:** 3
**Presentation:** 3
**Contribution:** 2
**Rating:** 4
**Confidence:** 4

**Summary:**

The paper addresses the subotimality of modern implicit neural representations (INR) with periodical activations (SIREN, FINER) related to suboptimal representational capacity. More specifically, it is observed that the architectures in question suffer from feature redundancy caused by a fixed frequency multiplier in the activation layer. The improvement proposed implies using adaptive frequency multiplier in the SIREN layers (Nyquist sampling rate determined by the layer width), which allow a more expressive representation learning. The paper then comprehensively compares the proposed change to the other periodical INRs, reporting mostly minor improvements on the image and surface overfitting and neural rendering experiments.

The overall proposal is simple and elegant in spirit, and introducing such altertantive easy-to-integrate initialisation to achieve minor yet consistent improvements across tasks is an attractive feature. However, the quantitative results achieved are rather minor and might not qualify for a top-tier conference contribution, especially since the experimental protocol for the experiments where the method brings most improvement is related to pure overfitting rather than true test-time performance. On the neural rendering task, the performance difference is negligible. This, combined with also some missing discussion of related work, contribute to the overall review leaning towards a reject. Fixing the aforementioned issues (improving quantitative results on unseen image pixels, stronger results in neural rendering and surface reconstruction experiments, extending the discussion to include most recent papers on INR optimisation) will contribute to a stronger paper and more positive outlook.

**Strengths:**

**S1. Periodic activation analysis and conceptual contribution.** The paper introduces a simple yet well-grounded idea for improving the efficiency and expressivity of implicit neural representations (INRs) through a principled initialization of periodic activation layers. The approach is conceptually clear and supported by solid motivation from sampling theory, offering an elegant bridge between signal processing intuition and neural representation learning.

**S2. Manuscript quality.** The paper is clearly written and well-organised. The exposition of the method, intuition, and experimental results is coherent and easy to follow, making the technical contribution accessible to a broad audience.

**Weaknesses:**

**W1. Weak quantitative results.** This is the primary weakness of the paper. To the best of my understanding, the main results are obtained on __overfitting__ experiments, where the model is trained and evaluated on the same signals, and its generalisation ability (for example, to unseen pixel locations) is not demonstrated. A fairer evaluation would follow the protocol of [1], where models are assessed on held-out subsets of pixel coordinates. More importantly, the results on a **practically relevant task** (NeRF) show only marginal improvements over the baseline and remain far from the state of the art. One way to strengthen the paper would be to demonstrate improved performance on a more challenging benchmark, such as the MipNeRF-360 [2] dataset and its corresponding baseline.

**W2. Missing related work.** The paper omits discussion of several closely related studies, notably BACON [3], FreSH [1] and TUNER [4]. FreSH addresses frequency allocation in implicit neural representations and introduces an improved strategy for selecting frequency ranges, supported by experiments that overlap in scope with the present work. More broadly, BACON [3] introduces coordinate networks with explicit frequency control enabling multi-scale scene representation. Finally, TUNER [4] focuses on stabilising the training of sinusoidal networks through adaptive frequency modulation to achieve better training stability. Including a comparison and discussion of these approaches would further clarify how the proposed method fits within and advances the recent literature on frequency-aware implicit representations.

**W3. Discussion of positional encoding and layer width could be expanded.**
The connection between the proposed method and positional encoding (PE) could be elaborated further, as FM-SIREN behaves similarly to PE in shallow networks. The authors note that PE requires selecting an appropriate encoding length L (lines 144–146), while in FM-SIREN the hidden-layer width implicitly defines the number of frequency multipliers and thus the spectral span. Although the paper includes an ablation on network width, extending this analysis to discuss its effect on expressivity and stability—and clarifying the analogy to PE—would strengthen the exposition and help readers better appreciate the design choices.


References:

[1] Kania, A., Mihajlovic, M., Prokudin, S., Tabor, J., & Spurek, P. (2024). FreSh: Frequency Shifting for Accelerated Neural Representation Learning. ICLR 2025.

[2] Barron, J. T., Mildenhall, B., Verbin, D., Srinivasan, P. P., & Hedman, P. (2022). Mip-nerf 360: Unbounded anti-aliased neural radiance fields. CVPR 2022.

[3] Lindell, David B., Dave Van Veen, Jeong Joon Park, and Gordon Wetzstein. "Bacon: Band-limited coordinate networks for multiscale scene representation." CVPR 2022.

[4] Novello, Tiago, Diana Aldana, Andre Araujo, and Luiz Velho. "Tuning the Frequencies: Robust Training for Sinusoidal Neural Networks." CVPR 2025.

**Questions:**

The main questions are:

**Q1**. The results on NeRF and other tasks suggest limited practical improvement. Could the authors clarify how the proposed method enhances **real-world*** applicability or generalisation (e.g., performance on unseen pixels or multi-view settings) beyond overfitting experiments? If faster convergence is one of the main advantages, could the authors also demonstrate similar benefits when training on real-world datasets, such as MipNeRF-360 [2], and its corresponding baseline model?

**Q2**. What are the key reasons for the **marginal gains** observed on the NeRF benchmark – do they stem from architectural constraints, limited spectral benefit, or experimental setup?

The authors are welcome to elaborate further on other points raised in the weaknesses section, such as a more direct comparison to FreSh (both conceptually and practically), and an expanded discussion of the relationship between positional encoding, layer width, and spectral span.

---

> ### Author Response · Authors · 2025-11-27
>
> We thank the reviewer for their feedback. We present a performance statement first. Then, we respond to weaknesses and questions in the same order.
>
> ### Performance Statement:
> We disagree with the characterization of our results as "minor." Our audio fitting achieves 84.6-90.8% MSE reduction (Table 1), an order of magnitude improvement over baselines. Image fitting demonstrates 2.4-7.6 dB PSNR gains (Table 2); in image compression standards, gains above 3 dB are considered substantial. We achieve 50% reduction in feature redundancy (Figure 2) and demonstrate single-layer networks outperforming classical DST (Figure 1)—bridging neural and signal processing methods. Moreover, our 3D shape fitting experiments demonstrate significant qualitative reconstruction while maintaining the same training time (Figure 5).
>
> **[W1]** We acknowledge that our primary evaluation focuses on complete signal reconstruction. However, this is the standard protocol for INR applications like neural compression and implicit storage, where perfect signal fitting is the goal. That said, we will add held-out coordinate experiments in our revision to demonstrate generalization. Our NeRF performance is better than what is presented by BACON [3] and FreSh [1], proving our models are also more robust in NeRF fitting.
>
> **[W2]** FreSH [1] is a search-based scheme for omega (ω) selection rather than a study of activation functions. FreSH selects a single frequency for the entire MLP, which suffers from feature redundancy as demonstrated in Figure 2. Our method assigns per-neuron frequencies to ensure orthogonality within layers; hence, it’s a fundamentally different approach. BACON [3] uses a different architecture than conventional MLPs, making direct comparison not feasible or even fair. Our method differs in maintaining standard MLP architecture while improving frequency diversity through principled initialization. TUNER [4] focuses on training stability through adaptive frequency modulation during optimization, whereas our method differs in assigning diverse frequencies to the fundamental combinatorial features of the MLP. We will include detailed discussion of these methods in our revised manuscript.
>
> **[W3]** PE is a ReLU-based MLP that encodes input coordinates into periodic embeddings. The main difference is that PE's encoding scheme is not informed by frequency limits, but rather uses uninformed frequency spacing defined by $2^{i(L-1)}$. This makes the ReLU-based MLP heavily tune the input frequencies while relying on non-convex optimization. Unlike PE, our FM-SIREN is pre-biased toward the true frequency range of the input signal, making it less prone to optimization difficulties. We will elaborate on this discussion in the related work section of our revised paper, as suggested by the reviewer.
>
> **[Q1]** We will include generalization experiments with held-out coordinates in our revision. Based on our improved performance, we conjecture that our approach should also benefit interpolation tasks.
>
> **[Q2]** NeRF fitting takes not only input coordinates but also viewing direction as input, which is not directly related to the Nyquist sampling theorem. We believe the modest performance inprovement is due to the weak relation to the Nyquist theorem.
>
> We thank the reviewer for insightful and helpful comments and suggestions.

---

### Official Review · Reviewer_JJrD · 2025-10-31

**Soundness:** 2
**Presentation:** 2
**Contribution:** 2
**Rating:** 4
**Confidence:** 4

**Summary:**

This paper addresses feature redundancy in implicit neural representations (INRs) that use periodic activations. Many INR uses the same activation function for all the layers, which limits the frequency diversity.This paper introduces FM-SIREN and FM-FINER, two new Nyquist-informed Implicit Neural Representation (INR) architectures designed to overcome redundancy in periodic activation networks such as SIREN and FINER.  Experiment result on 1D audio, 2D image, 3D shape, and NeRF synthesis, showing consistent performance gains and efficiency improvements.

**Strengths:**

This introduces frequency diversity without requiring hyperparameter tuning or additional network depth. It achieves a 49.92% and 50.43% improvement in the Frobenius norm of the covariance matrix compared to their baselines. It also shows a strong performance in reconstruction on the Kodak and BSDS500 datasets.

**Weaknesses:**

[1] The Discrete Sine Transform (DST) assumes the odd symmetry at the boundaries: How do you confirm that these datasets satisfy these assumptions?

[2] The Nyquist Sampling Theorem requires uniform sampling. However, INRs don’t use uniformly sampled signals in the classical sense. Instead, they approximate a continuous mapping. Even though the training data might be uniformly spaced in a dataset (e.g., pixels or audio samples), the INR’s internal representation and evaluation grid can be nonuniform or adaptive depending on optimization dynamics, loss weighting, or coordinate encodings. How do you make sure that Nyquist Sampling Theorem can apply well to INR?

[3] The method required to apply in the frequency domain, can it be applied to the non-frequency domain?

[4] Even though the dataset covers from 1D to 3D. How is its performance in the PDE dataset?

[5] More recent baselines should be included. For example, VI^3NR [r1], FINER++ [r2]

[r1]Koneputugodage, Chamin Hewa, et al. "VI^ 3NR: Variance Informed Initialization for Implicit Neural Representations." Proceedings of the Computer Vision and Pattern Recognition Conference. 2025
[r2] Zhu, Hao, et al. "FINER++: Building a Family of Variable-periodic Functions for Activating Implicit Neural Representation." arXiv preprint arXiv:2407.19434 (2024).

**Questions:**

See the Weakness

---

> ### Author Response · Authors · 2025-11-27
>
> ## Part 1: Clarifying Fundamental Concepts
> We thank the reviewer for his/her feedback. Before addressing the specific weaknesses mentioned by the reviewer, it might be helpful to clarify some fundamental points about our approach, as these affect the interpretations of our contributions:
>
> ### Clarification #1: Frequency domain vs. Coordinate domain
> The reviewer states that our method only applies "in the frequency domain," but this is not how INRs work. We emphasize that all INRs, including SIREN, FINER, and our approach, operate in the coordinate/spatial domain (e.g., mapping pixel coordinates to RGB values, or 3D positions to density/color in NeRF). The term "frequency" in our context refers to the frequency of the periodic activation functions (which are sinusoids), not the data representation domain. All our experiments are in coordinate space: images (2D spatial), audio (1D temporal), 3D shapes (3D spatial), and NeRF (3D spatial + 2D viewing direction). There is no "frequency domain" assumption as such.
>
> ### Clarification #2: DST and odd symmetry
> The reviewer believes we require input data to satisfy DST's odd symmetry property. This is not the case. We use DST as inspiration for a design principle. We basically assign distinct frequencies to basis functions to ensure orthogonality which reduces redundancy. Consider the following flow:
>
> 1- Classical DST: Uses sine functions as bases with different frequencies (orthogonal basis functions)
> 2- SIREN: Uses the same frequency (ω₀) for all neurons, leading to redundancy as shown in Fig. 2a
> 3- FM-SIREN (Our method): Uses different frequencies per neuron, leading to orthogonal activations and reduced redundancy as shown in Fig. 2c
> 4- The same applies for FINER and FM-FINER
>
>
> ### Clarification #3: Nyquist theorem application
> The reviewer seems to conflate the following:
> 1- Input sampling (how densely we sample the data)
> 2- Frequency allocation (what frequencies the activation functions can represent)
>
> Given the maximum frequency present in a signal is $f_{max}$, sampling frequency should be at least $2f_{max}$ or higher. We use this to inform the range of frequency multipliers for our activation functions (Eq. 6), noting that f_s/2 is obtainable from the number of samples given in a signal. This ensures that neurons collectively cover the representable frequency spectrum without aliasing.
>
> ## Part 2: Addressing Specific Weaknesses
> Now that we've established this common understanding, we address each weakness:
>
> ### [W1] DST and odd symmetry at boundaries
> Our method does not assume or require odd symmetry in the data. The key principle is frequency diversity among neurons, not data symmetry. We have proven that our results across different and diverse datasets (Kodak, BSDS500, Stanford 3D, Blender) with different characteristics demonstrate this generality.
>
> ### [W2] Nyquist Sampling and uniform spacing
> As the Nyquist theorem defines the maximum frequency present in a signal given the sampling rate (or vice versa), When a signal is sampled non-uniformly, the maximum frequency present is still dependent on the maximum sampling rate ($max( \frac{f_s}{2} ))$. Therefore, Nyquist theorem still holds for non-uniform sampling considering the higher sampling rate used in a signal. This is still applicable to our methods as it will inform the frequency multiplier range ($ω_k$) of the MLP’s layers. Thereafter, the resultant MLP can reconstruct the signal at any coordinate the space since it is a continuous function, with uniform or non-uniform sampling.
>
> ### [W3] Frequency vs. non-frequency domains
> The term "frequency domain" does not apply to any of our experiments as clarified above.
>
> ### [W5] Missing recent baselines
> [r1] Focuses on variance-preserving initialization for SIREN which is orthogonal to our contribution (activation frequency allocation). We can add this comparison if the reviewer believes it's essential, but we note here that it addresses a very different problem (initialization vs. activation design)

---

### Meta-Review · Area_Chair_6MGd · 2025-12-23

**Summary:**

All reviewers raised major concerns about this manuscript and provided overall negative evaluations. While the authors have addressed some minor comments, the major concerns remain unresolved. In particular, issues related to missing references, positional encoding, and comparison with more recent methods have not been adequately addressed.

**Reviewer Concerns:**

Minor concerns regarding the Nyquist sampling theorem were addressed.

**Reviewer Scores:**

The primary limitation of this paper is the absence of comparisons with recent state-of-the-art methods. In particular, several relevant works published within the past year are entirely omitted. Additionally, the experimental results are reported only on hand-picked examples. Demonstrating strong performance on a small number of selected images does not provide sufficient evidence that the method generalizes across an entire dataset.

A more convincing evaluation would include quantitative results on all images in a dataset, for example using summary visualizations such as radar plots, along with direct comparisons to existing approaches. It is also unclear how the proposed method performs relative to other recent methods in the literature, and performance should be compared to relevant works, for example:

- D. Jayasundara, H. Zhao, D. Labate, V. M. Patel, MIRE: Matched Implicit Neural Representations, IEEE/CVF CVPR 2025.
- K. Vyas, A. I. Humayun, A. Dashpute, R. Baraniuk, A. Veeraraghavan, G. Balakrishnan, Learning Transferable Features for Implicit Neural Representations, NeurIPS 2024.

---

### Decision · Program_Chairs · 2026-01-26

Reject